# Evaluating the Role of Great Pre-trained Diffusion Models in Few-shot Phase: Warm-up and Acceleration

## Abstract

Due to the customized requirements, few-shot diffusion models have attracted much attention. Despite the empirical success, only a few works analyze few-shot models, and they do not involve the fast few-shot optimization process. However, fast optimization is important and necessary in quickly responding to users. In this work, for the first time, we evaluate the role of each operation in the optimization process and prove the convergence guarantee for few-shot diffusion models. A standard operation for the few-shot model is only fine-tuning some key parameters to avoid overfitting the limited target dataset. We first show that this operation is insufficient from empirical and theoretical perspectives. More specifically, we conduct real-world few-shot fine-tuning experiments with underfitting and overfitting bad pre-trained models and show that the few-shot results are heavily influenced by these bad models. Theoretically, we also prove that the few-shot phase can not learn the ground-truth parameters and suffers a small gradient when using a bad pre-trained model. Based on these observations and theoretical guarantees, we highlight the importance of a great pre-trained model by showing it can warm up few-shot models and lead to a strongly convex landscape for few-shot diffusion models. As a result, the few-shot model fast converges to the ground-truth parameters. In contrast, we show that with a bad initialization, the pretraining phase requires large optimization steps to converge. Combined with the above results, we explain why few-shot diffusion models only require a few optimization steps compared with the pretraining phase.

## 1 Introduction

Recently, diffusion models, which are trained on large-scale datasets with sufficient training time, have shown impressive performance in different areas such as 2D and 3D generation (Rombach et al., 2022; Blattmann et al., 2023; Liu et al., 2024). However, when facing customized requirements, we only have limited data and need to achieve a quick and high-quality response to users. To achieve great performance under such a situation, few-shot diffusion models have received attention (Ruiz et al., 2023; Xiang et al., 2023; Kumari et al., 2023; Moon et al., 2022; Liu et al., 2023). Few-shot diffusion models only use a limited target dataset ($5 - 10$ images) and a few optimization steps (fewer than $1k$ steps) to fine-tune a pre-trained model (such as Stable Diffusion (SD) XL, which requires $500k$ optimization steps) and generate samples with the target feature.

Though few-shot diffusion models achieve great performance in applications, only a few works aim to explain the success of few-shot diffusion models (Yang et al., 2024a; Chua et al., 2021; Cheng et al., 2025). Furthermore, they focus on the estimation error and explain why a limited target dataset is enough for few-shot models. However, the fast optimization process of few-shot diffusion models is also important and necessary for quick response to users, and the theoretical guarantee for it is lacking. Hence, the following natural question remains open:

*Why do few-shot diffusion models only require a few optimization steps to achieve great performance?*

In this work, for the first time, we study the optimization process of few-shot diffusion models, highlight the role of great pre-trained models, and answer the above question by showing that the few-shot phase fast converges to the ground-truth parameters under a suitable condition.

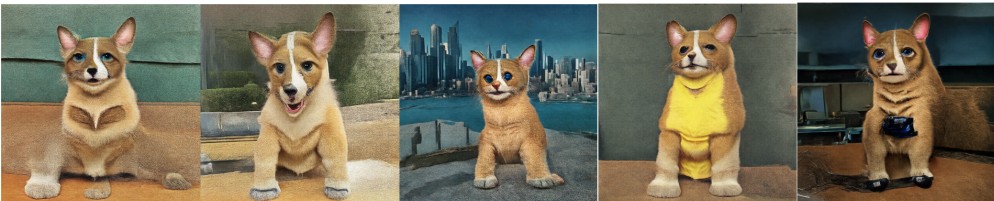

Fine-tuning Results based on Great Pre-trained Models (SD3 Medium)

Fine-tuning Results based on *Overfitting* Bad Pre-trained
Models (SD3 Medium with 1k overfitting steps)

*A cat on top of a*          *A cat in a chef*          *A cat with a city*          *A cat wearing a*          *A cat in a police*
*wooden floor*               *outfit*                  *in the background*         *yellow shirt*             *outfit*

Figure 1: DreamBooth few-shot fine-tuning results based on great and bad *overfitting* pre-trained
Models. The overfitting bad pre-trained Model is obtained by training SD3 Medium with 5 dog
image for $1k$ steps. The generated images based on overfitting bad pre-trained models suffer from the
memory phenomenon of the overfitting feature.

Before providing the convergence guarantee, we first analyze the necessary conditions for a great
few-shot diffusion model. A standard operation for few-shot diffusion models is to freeze most
parameters and only fine-tune some key parameters (Liu et al., 2023; Xiang et al., 2023). However,
we show that this operation is not enough from the empirical and theoretical perspectives. More
specifically, we conduct real-world experiments and show that with bad pre-trained models, the
few-shot phase can not generate high-quality images, where overfitting bad pre-trained models suffer
from the memory phenomenon (Figure 1) and underfitting bad pre-trained models have a fine-tuning
loss gap (Figure 2). Based on our experiment observation, we prove that if the pre-trained model is
bad, few-shot diffusion models can not learn the ground-truth parameters. Furthermore, the gradient
of few-shot diffusion models becomes small when the point is still far away from the minimizer.
In other words, under this setting, few-shot diffusion models require large optimization steps to
converge. As a byproduct of the gradient analysis, we also show that the pretraining phase with a bad
initialization suffers from a small gradient, which slows down the optimization process.

The above results can not explain why few-shot diffusion models can only use a few optimization
steps to achieve great performance. Based on the analysis of bad pre-trained models, we show the
importance of great pre-trained models. An intuition is that great pre-trained models provide a
warm-up for the few-shot phase and simplify the landscape. Inspired by this intuition, we prove that
the few-shot phase with a great pre-trained diffusion model converges to the ground truth parameters
using the gradient descent algorithm and provide a convergence guarantee for the optimization
process.

Combined with the analysis for the pretraining phase, these results explain why few-shot models can
use much smaller optimization steps to achieve great performance. In conclusion, for the first time,
we analyze the optimization process of few-shot diffusion models and achieve the following results:

- By providing real-world experiments and counter-examples, we prove that a great pre-trained
  model plays an important role in the few-shot phase. Otherwise, few-shot diffusion models
  can not learn the ground-truth parameters and require large optimization steps to converge.

- We show that with a great pre-trained model, the landscape of few-shot diffusion models
  becomes strongly convex. As a result, few-shot models quickly converge to the ground-truth
  parameters, and we prove the convergence guarantee for this optimization process.

- As a byproduct of gradient analysis, we prove that the pretraining phase with a bad initial-
  ization suffers a small gradient and requires large optimization steps.

## 2 RELATED WORK

**Optimization Guarantee for Diffusion Models.** Since the score matching objective function is non-convex, only a few works analyze the optimization process of diffusion models. Furthermore, these works either focus on some specific data distribution or use the kernel method to simplify the analysis. For the specific data distributions, a series of works focus on designing algorithms to learn Gaussian Mixture Models (Bruno et al., 2023; Cui and Zdeborová, 2023; Shah et al., 2023; Chen et al., 2024) based on the score matching technique. Han et al. (2024a) focus on a data distribution consisting of two fixed orthogonal vectors. For the general data, Li et al. (2023) and Han et al. (2024b) simplify the problem to a convex optimization by using a wide 2-layer NN and kernel methods. Then, they use the gradient descent (flow) method to obtain a convergence guarantee. We note that the above works focus on the pretraining phase. On the contrary, we focus on the few-shot phase, discuss the necessary conditions, and provide the first convergence guarantee for few-shot diffusion models.

**Guarantee for Few-shot Diffusion Models.** Recently, some works have focused on the estimation error of few-shot diffusion models (Yang et al., 2024a; Chua et al., 2021; Cheng et al., 2025). The core idea is to model the shared part between the source (meta learning phase) data and the target data. Based on this intuition, Yang et al. (2024a) show that few-shot diffusion models escape the curse of dimensionality and make the first step to explain the empirical success of few-shot diffusion models. Recently, Chua et al. (2021) and Cheng et al. (2025) study the conditional diffusion models and also prove the estimation error of the few-shot phase with the meta-learning prior information. Only Yang et al. (2024a) study the optimization process and provide a closed-form minimizer for the linear subspace distribution with a Gaussian latent. However, the real-world distribution is always multi-modal, and diffusion models usually use optimization algorithms instead of obtaining the closed-form minimizer. Hence, in this work, we focus on the multi-modal latent distribution and use the gradient descent (GD) algorithm to optimize the objective function.

## 3 PRELIMINARIES

We first introduce the basic knowledge and notation of diffusion models. Let $q_0 \in \mathbb{R}^D$ be the data distribution. The variance preserving (VP) forward process is defined by:

$$\mathrm{d}x_t = -x_t\mathrm{d}t + \sqrt{2}\mathrm{d}B_t, x_0 \sim q_0 \in \mathbb{R}^D ,$$

where $\{B_t\}_{t \in [0,T]}$ is a $D$-dimensional Brownian motion. Let $q_t$ be the density function of $x_t$ and $\{y_t\}_{t \in [0,T]} = \{x_{T-t}\}_{t \in [0,T]}$. To generate samples, diffusion models reverse the forward process and run the corresponding reverse process:

$$\mathrm{d}y_t = [y_t + 2\nabla \log q_{T-t}(y_t)] \, \mathrm{d}t + \sqrt{2}\mathrm{d}B_t .$$

The reverse process requires the score function $\nabla \log q_t(\cdot)$, which contains the data information and can not be exactly calculated. A conceptual way to approximate $\nabla \log q_t(\cdot)$ is to minimize the following score matching (SM) objective function (Song et al., 2020; Karras et al., 2022):

$$\min_{s \in \mathrm{NN}} \mathcal{L}_{\mathrm{SM}} = \int_\delta^T \mathbb{E}_{x_t \sim q_t} \|\nabla \log q_t(x_t) - s(x_t,t)\|_2^2 \, \mathrm{d}t , \tag{1}$$

where NN is a given function class and $\delta > 0$ is the early stopping parameter to avoid a blow-up score. However, $\mathcal{L}_{\mathrm{SM}}$ can not be directly calculated since $\nabla \log q_t(\cdot)$ is unknown for general data. To avoid this problem, Vincent (2011) propose the denoising score matching (DSM) loss based on the conditional score function $\nabla \log q_t(x_t|x_0)$ with an analytical form:

$$\min_{s \in \mathrm{NN}} \mathcal{L}_{\mathrm{DSM}} = \int_\delta^T \mathbb{E}_{x_0} \left[ \mathbb{E}_{x_t|x_0} \|\nabla \log q_t(x_t|x_0) - s(x_t,t)\|_2^2 \right] \mathrm{d}t ,$$

which is equivalent to $\mathcal{L}_{\mathrm{SM}}$ up to a constant independent of the optimized parameters. Once a forward process is chosen, $q_t(x_t|x_0)$ is determined as $q_t(x_t|x_0) = \mathcal{N}(m_t x_0, \sigma_t^2 I_D)$, and $\nabla \log q_t(x_t|x_0)$ has an analytical form $-(x_t - m_t x_0)/\sigma_t^2$, where $m_t = e^{-t}, \sigma_t^2 = 1 - m_t^2$ for VP forward process.

With a score, diffusion models discretize and run the reverse process to generate samples. Since the sampling process is widely studied (Chen et al., 2022; Yang et al., 2024b) and the optimization analysis is lacking due to the highly nonlinear score, this work focuses on the optimization process.

### 3.1 Two Phases of Few-shot Diffusion Models

After discussing diffusion models, we introduce few-shot diffusion models widely used in applications (Kumari et al., 2023; Moon et al., 2022; Liu et al., 2023), which consist of two phases: the pretraining phase and the few-shot phase. In the pretraining phase, we train a diffusion model with a large source dataset and sufficient optimization steps. In the few-shot phase, we freeze most parameters and fine-tune some key parameters corresponding to target features with a limited target dataset.

As a beginning, we introduce assumptions on data. Following Yang et al. (2024a), we assume source distribution $q_s$ and target distribution $q_{ta}$ both admit linear subspaces and share a latent distribution.

**Assumption 3.1.** Source data $x_s$ and target data $x_{ta}$ have form $x_s = A_s z$ and $x_{ta} = A_{ta} z$ where $A_s, A_{ta} \in \mathbb{R}^{D \times d}$ have orthonormal columns and $z \sim q_z \in \mathbb{R}^d$.

The low-dimensional structure has been discovered in image and text datasets (Pope et al., 2021; Tenenbaum et al., 2000), and the linear subspace assumption has been widely adopted in many previous theoretical works (Chen et al., 2023; Yuan et al., 2023; Guo et al., 2024). For the shared latent assumption, it is used by current analysis for few-shot diffusion models (Yang et al., 2024a) and is standard in the context of few-shot learning (Du et al., 2020; Meunier et al., 2023). With the linear subspace assumption, the score function can be decomposed into (1) a latent score $\nabla \log q_t^{\mathrm{LD}}(\cdot)$ and (2) linear encoder and decoder $A_s$ ($A_{ta}$ for target distribution) (Chen et al., 2023)

$$\nabla \log q_t^s(x) = A_s \nabla \log q_t^{\mathrm{LD}}(A_s^\top x) - (I_D - A_s A_s^\top)x/\sigma_t^2 ,$$

where $q_t^{\mathrm{LD}}(z') = \int q_t(z'|Z) q_z(z) \mathrm{d}Z$ and $q_t(\cdot|z) = \mathcal{N}(m_t z, \sigma_t^2 I_d)$. This decomposition means that the optimization process needs to optimize two parts: the linear encoder and decoder $A_s$ (parameterized by $V_s$; $A_{ta}$ parameterized by $V_{ta}$) and latent score $\nabla \log q_t^{\mathrm{LD}}(\cdot)$ (parameterized by $\mu$). Then, the objective function for the pretraining phase is

$$\min_{s \in \mathcal{S}_{\mathrm{NN}}} \mathcal{L}_{\mathrm{DSM}}^{\mathrm{pre}} = \int_\delta^T \mathbb{E}_{x_0 \sim q_s} \left[ \mathbb{E}_{x_t|x_0} \|\nabla \log q_t^s(x_t|x_0) - s(x_t, t)\|_2^2 \right] \mathrm{d}t ,$$

where $\mathcal{S}_{\mathrm{NN}}$ is the function class used in the pretraining phase and has the following form

$$\mathcal{S}_{\mathrm{NN}} = \big\{ \mathbf{s}_{V,\mu}(x, t) = V\mathbf{f}_\mu(V^\top x, t)/\sigma_t^2 - x/\sigma_t^2 : V \in \mathbb{R}^{D \times d} \text{ with orthonormal columns,}$$
$$\mathbf{f}_\mu : \mathbb{R}^d \times [\delta, T] \to \mathbb{R}^d \text{ a network} \big\}.$$

With a pre-trained score function, the diffusion model fine-tunes it with a given target dataset in the few-shot phase. Let $(\widehat{V}_s, \hat{\mu})$ be the minimizer of the above pretraining objective function. Since the source and target data share a latent distribution, we freeze the approximated latent score function $\mathbf{f}_{\hat{\mu}}$ and only fine-tune the linear encoder and decoder $V_{ta}$ in the fine-tuning phase:

$$\min_{s \in \mathcal{Q}_{NN}(\hat{\mu})} \mathcal{L}_{\mathrm{DSM}}^{\mathrm{few}} = \int_\delta^T \mathbb{E}_{x_0 \sim q_{ta}} \left[ \mathbb{E}_{x_t|x_0} \|\nabla \log q_t^{ta}(x_t|x_0) - s(x_t, t)\|_2^2 \right] \mathrm{d}t , \qquad (2)$$

where $\mathcal{Q}_{\mathrm{NN}}(\mu) = \{\mathbf{s}_{V,\mu}(x, t) = \frac{1}{\sigma_t^2} V\mathbf{f}_\mu(V^\top x, t) - \frac{1}{\sigma_t^2} x : V \in \mathbb{R}^{D \times d} \text{ with orthonormal columns.}\}$.

**Notations.** We denote by $I_D$ the $D$-dimensional identity matrix and $\mathbf{I}$ the matrix with all elements equal to 1. For a vector $x \in \mathbb{R}^D$, we denote by $\|x\|_2$ the Euclidean norm and $x(i)$ the $i$-th element. For a matrix $A \in \mathbb{R}^{D \times d}$, we denote by $\|A\|_F$ the Frobenius norm and $A(i, j)$ the $(i, j)$-th element. For the optimization, we define $z^{(0)}$ by the initialization and $z^{(k)}$ the $k$-th iteration of GD algorithm.

To characterize the landscape of the objective function, we give the following definition.

**Definition 3.2.** $\phi : \mathbb{R}^D \to \mathbb{R}$ is $\lambda$-strongly convex and $L_m$-smooth if $\lambda I_D \preceq \nabla^2 \phi(x) \preceq L_m I_D$.

## 4 The Influence of Bad Pre-trained Models in Few-shot Phase

As a start, we conduct experiments to show the influence of bad pre-trained models in the few-shot phase. The bad pre-trained models can be roughly divided into overfitting and underfitting, where the former suffers from low diversity and the latter does not learn the basic information.

**Overfitting Bad Pretraining: Memory Phenomenon.** Since Stable Diffusion (SD) models can generate high-quality and diverse samples, we view them as great pretrained models. To obtain overfitting

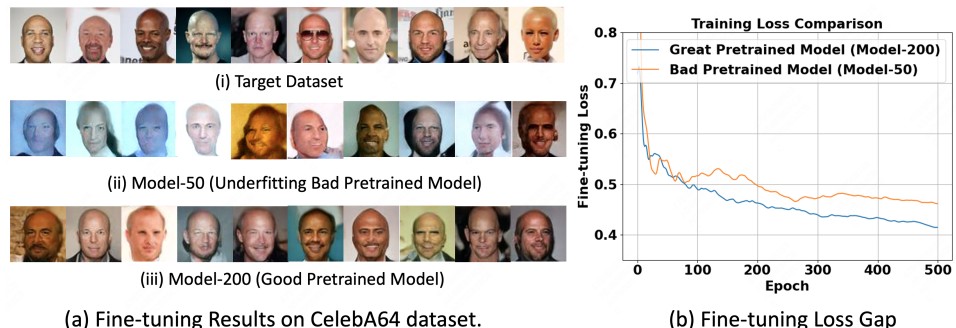

(i) Target Dataset

(ii) Model-50 (Underfitting Bad Pretrained Model)

(iii) Model-200 (Good Pretrained Model)

(a) Fine-tuning Results on CelebA64 dataset.

(b) Fine-tuning Loss Gap

Figure 2: The *underfitting* experiments on CelebA64 dataset. Based on the underfitting bad pretrained models, the few-shot phase can not generate clean face images and suffers from the loss gap.

bad models, we overfit them with one prompt (a photo of a dog) and 5 dog images. As the overfitting step increases, the diversity decreases. We use two bad pretrained models, the first one overfits 1k steps (Bad pretrain1k, Bad1k), and the latter, a worse one, overfits 4k steps (Bad pretrain4k, Bad4k).

In the few-shot phase, following Dreambooth (Ruiz et al., 2023), we fine-tune the pre-trained model with Dreambooth training dataset. For the evaluation, we use Dreambooth test prompts and generate $3k$ images to calculate Clip-T and Pickscore. To match Section 3.1, we only fine-tune the first and last 3 blocks of NN (Details in Appendix C).

|  |  | Great | Bad1k | Bad4k |
|---|---|---|---|---|
| SD1.4 | Clip-T | **0.3258** | 0.3240 | 0.3227 |
|  | Pickscore | **21.70** | **21.70** | 21.67 |
| SD3-M | Clip-T | **0.3241** | 0.3085 | 0.2357 |
|  | Pickscore | **22.11** | 20.52 | 18.52 |

Table 1: Results for overfitting bad pretraining.

As shown in Figure 1, if the pre-trained models overfit a dog feature, the few-shot phase suffers from the memory phenomenon and can not generate images with the target feature, for example, a cat feature. Table 1 also shows that as the pretrained models become worse, the metric is worse. Another interesting observation is the difference between the SD1.4 and SD3-M. As shown in Table 1, SD3-M suffers a heavy influence of bad pretrained models compared to SD1.4. One core difference is that SD 3 adopts the deterministic sample process, and SD1.4 adopts the stochastic sample process. This indicates the learning error in the fine-tuning phase (introduced by the bad pretrained models) quickly accumulates through the sampling process when adopting the deterministic sampler.

**Underfitting Bad Pretraining: Few-shot Loss Gap.** Following Yang et al. (2024a), we conduct experiments on CelebA64 to show the influence of underfitting pre-trained models. We first train two basic models with different hairstyles. The facial features generated by the basic model trained with 50 epochs (Model-50) are distorted, while Model-200 (with converged loss) can generate clear faces. Hence, we call Model-50 an underfitting model and Model-200 a great pre-trained model.

Then, we fine-tune the appropriate encoder and decoder (See details in Appendix C) of the two basic models with 10 bald hairstyle target images. As shown in Fig. 2 (a), the Model-50 usually generates images with distorted facial features after fine-tuning, which is due to the poor learning of basic concepts (such as every face has a nose, eyes, etc.) in the pretraining phase. On the contrary, the Model-200, after fine-tuning, can generate novel images with the target bald feature. Fig. 2 (b) shows the few-shot fine-tuning loss gap between the great and underfitting bad pre-trained models. This loss gap indicates that if the basic feature and concepts are not learned in the pretrianing phase, it is hard to make up during the few-shot fine-tuning phase and leads to a bad local minima.

In the following part, we provide the theoretical explanation for the few-shot loss gap and the gradient analysis for the few-shot phase with a bad pre-trained model, which indicates that a bad pre-trained model can not warm up and is hard to provide a good initialization for the few-shot phase.

## 5 BAD PRETRAINING PREVENTS FEW-SHOT PHASE LEARNING PARAMETERS

An intuitive idea is that the few-shot phase will quickly converge to the global minimizer since it only optimizes fewer parameters than the pretraining phase. However, we show that if the pre-trained

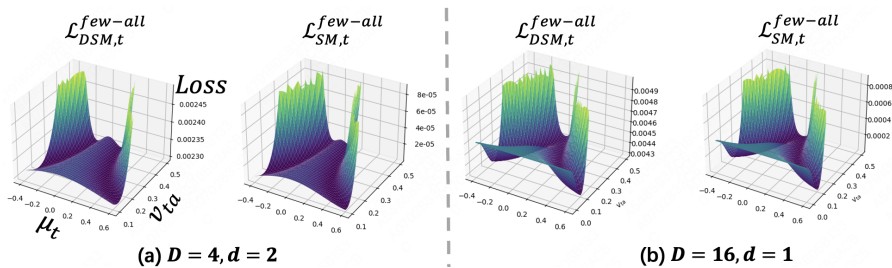

Figure 3: The Landscape for $\mathcal{L}_{\text{SM}}$ and $\mathcal{L}_{\text{DSM}}$. Since the landscape of $\mathcal{L}^{\text{few}}$ can be viewed as a slice of $\mathcal{L}^{\text{few}-\text{all}}$ at $\hat{\mu}$, we present the landscape of $\mathcal{L}^{\text{few}-\text{all}}$.

model is bad (the criteria of great pretrained models is provided in Section 6.1), the few-shot phase can not learn $A_{ta}$, suffers a constant error (few-shot loss gap), and has a small gradient.

Before providing our results, we first make some assumptions on the latent distribution $q_z$. As discussed in Sec.2, Yang et al. (2024a) only adopt a Gaussian latent, which can not reflect the multi-modal property of the real-world dataset. On the contrary, we assume the latent distribution is a mixture of Gaussians, which is multi-modal and has a nonlinear score function.

**Assumption 5.1.** The external dimension $D = 2$ and latent dimension $d = 1$. The latent distribution is $q_z = \frac{1}{2}\mathcal{N}(-\mu^*, 1) + \frac{1}{2}\mathcal{N}(\mu^*, 1)$, and the linear parts are $A_s = [a_s, a_s]^\top$ and $A_{ta} = [a_{ta}, a_{ta}]^\top$.

*Remark* 5.2. The 2-modal Gaussian mixture distribution with symmetrical mean and standard variance is also used in Shah et al. (2023), and this latent distribution is representative since it has multi-modal properties and a nonlinear score function. We note that the assumption can be extended to general $D$ and $d$, and the toy case with $D = 2$ and $d = 1$ is used for the convenience of calculation. The simulation experiments also support our discussion (Table 2), and we also provide some promising methods to extend this assumption to general latent distribution and general manifold (Section 6.1).

Let $\mu_t^* = \mu^* \exp(-t)$. After assuming 2-modal Gaussian Mixture latent, the ground truth latent score $\nabla \log q_t^{\text{LD}}(\cdot)$ has a closed form, which leads to the following score in the full space:

$$\nabla \log q_t(x) = A \tanh(\mu_t^{*\top} A^\top x)\mu_t^* - AA^\top x - (I_D - AA^\top)x/\sigma_t^2 . \tag{3}$$

Inspired by the above formulation, we use the following network $\mathbf{f}_\mu(z, t)$ to approximate latent score:

$$\mathbf{f}_\mu(z, t) = \sigma_t^2 \tanh(\mu_t^\top z)\mu_t + (1 - \sigma_t^2)z ,$$

and $V_{ta} = [v_{ta}, v_{ta}]^\top$. After determining $\mathcal{S}_{\text{NN}}$ and $\mathcal{Q}_{\text{NN}}$, the few-shot diffusion models can first optimize $\mathcal{L}_{\text{DSM}}^{\text{pre}}$ and fine-tune the pre-trained score with $\mathcal{L}_{\text{DSM}}^{\text{few}}$ with the target dataset. We also define the score matching objective function for the few-shot phase, which is used in the analysis:

$$\min_{s \in \mathcal{Q}_{\text{NN}}(\hat{\mu})} \mathcal{L}_{\text{SM}}^{\text{few}} = \int_\delta^T \mathbb{E}_{X_t \sim q_t^{ta}} \|\nabla \log q_t^{ta}(x_t) - s(x_t, t)\|_2^2 \, dt .$$

We note that $\mathcal{L}_{\text{SM}}^{\text{few}}$ and $\mathcal{L}_{\text{DSM}}^{\text{few}}$ are equivalent up to a constant independent of all optimized parameters (Vincent, 2011), which indicates the optimization landscape is the same for these objective functions (Fig. 3). Since the score $\nabla \log q_t(\cdot)$ under Assumption 3.1 and 5.1 has an analytical form (Eq. 3), we focus on the score matching (SM) objective function. We note that when considering the convergence guarantee of the pretraining phase, Li et al. (2023) also adopt the score matching objective $\mathcal{L}_{\text{SM}}^{\text{pre}}$.

## 5.1 RESULTS FOR FEW-SHOT MODELS WITH A BAD PRETRAINING

Since linear matrix $A$ is independent of $t$, we fix a $t \in [\delta, T]$ and use the objective function $\mathcal{L}_{\text{SM},t}^{\text{few}}$ in this work (the influence of $t$ is discussed in Remark 6.5). For convenience, we call the pre-trained model great when the latent parameter $\hat{\mu} = \mu^*$. Otherwise, we call the pre-trained model bad [1]. We note that the underfitting and overfitting pre-trained models satisfy this definition, where the former do not learn basic concepts and the latter overfit to some specific features.

---

[1]Since the source dataset are limited, $\|\hat{\mu} - \mu^*\|$ is smaller than a small constant $\epsilon_{\text{pre}}$ instead of equal to 0 for a great pre-trained model. In Section 6.1, we discuss the influence of limited data and imperfect learning.

Let $\widehat{V}_{ta}$ be the solution of $\partial \mathcal{L}_{\mathrm{SM},t}^{\mathrm{few}}/\partial V_{ta} = 0$. In this part, we show that with a bad pre-trained model, $\|\widehat{V}_{ta}\widehat{V}_{ta}^\top - A_{ta}A_{ta}^\top\|_F$ is not equal to 0, which indicates the few-shot phase can not learn ground-truth subspace parameters and suffers a few-shot loss gap.

**Lemma 5.3.** *Assume Assumption 3.1 and 5.1 . If $\hat\mu \neq \mu^*$, with $V_{ta}V_{ta}^\top = A_{ta}A_{ta}^\top$, $\partial \mathcal{L}_{\mathrm{SM},t}^{\mathrm{few}}/\partial V_{ta} \neq 0$.*

This lemma indicates that $\|\widehat{V}_{ta}\widehat{V}_{ta}^\top - A_{ta}A_{ta}^\top\|_F \neq 0$ if $\hat\mu \neq \mu^*$, which explain the few-shot loss gap in Fig.2. Then, we discuss the influence of $|\hat\mu - \mu^*|$ by using a simplified example $\hat\mu = 0$ and $\mu^* \neq 0$.

**Theorem 5.4.** *Assume Assumption 3.1 and 5.1 hold. Let $\mu_1^*$ and $\mu_2^*$ be the two parameters to generate different latent distributions. Given a bad pre-trained model with $\hat\mu = 0$, if $|\mu_1^* - \hat\mu| > |\mu_2^* - \hat\mu|$, then*

$$\|\widehat{V}_{ta,1}\widehat{V}_{ta,1}^\top - A_{ta}A_{ta}^\top\|_F > \|\widehat{V}_{ta,2}\widehat{V}_{ta,2}^\top - A_{ta}A_{ta}^\top\|_F,$$

*where $\widehat{V}_{ta,i}$ is the solution corresponds to $\mu_i^*, i \in \{1,2\}$.*

This result shows that with a worse pre-trained model, the solution of the few-shot phase becomes worse. Hence, a great pre-trained model is necessary for the few-shot phase. Before providing positive results, we further prove that with a bad pre-trained model, another fully fine-tuning method for few-shot models also has a bad performance and suffers from a small gradient.

**Fully Fine-tuning Method and Gradient Analysis.** Though many empirical works only fine-tune key parameters (Liu et al., 2023; Kumari et al., 2023; Moon et al., 2022), a few work (Ruiz et al., 2023) still optimize all parameters $\min_{s \in \mathcal{S}_{\mathrm{NN}}} \mathcal{L}_{\mathrm{SM}}^{\mathrm{few-all}}$ in the few-shot phase with initialization $(\widehat{V}_s, \hat\mu)$, where $\mathcal{L}_{\mathrm{SM}}^{\mathrm{few-all}}$ is the same with Eq. 1. In the following theorem, we show that with a bad pre-trained model, $\partial \mathcal{L}_{\mathrm{SM},t}^{\mathrm{few-all}}/\partial \mu_t$ is small when the point is far away from the global minimizer.

**Theorem 5.5.** *Assume Assumption 3.1 and 5.1 holds. For a fixed $t$, if $\mu_t \in (-\epsilon, \epsilon)$, we have that*

$$\partial \mathcal{L}_{\mathrm{SM},t}^{\mathrm{few-all}}/\partial \mu_t \leq 4\epsilon A_{ta}^\top V_{ta}\sqrt{(1 + \mu_t^{\star 2})\, V_{ta}^\top V_{ta}}\sqrt{C_1} + O(\epsilon^{\frac{3}{2}}),$$

*where $C_1$ is a small constant determined by $V_{ta}$, $A_{ta}$ and $\mu^*$ (Details in Eq. 10).*

The above result indicates that if $\hat\mu_t \in (-\epsilon, \epsilon)$, the gradient is small. Then, if $\mu^*$ is a positive constant larger than $\epsilon$, the few-shot phase requires large optimization steps to get rid of the bad pretraining phase. We also use a toy example to show the scale of the gradient, which is much smaller than $\epsilon$.

**Example 5.6.** *Considering $A_s = [0.1, 0.1]^\top$, $A_{ta} = [0.12, 0.12]^\top$, and $\mu^* = 4$. With a fixed $t = 2$ and $V_{ta} = [0.1, 0.1]$ (close to the $A_{ta}$), $\partial \mathcal{L}_{\mathrm{SM},t}^{\mathrm{few-all}}/\partial \mu_t \leq 1 \times 10^{-5}$ when $\mu_t \in (-0.12, 0.12)$.*

*Remark* 5.7 (Limited Target Data). Theorem 5.5 consider the gradient of the fully fine-tuning method in the exception. When considering a limited target dataset, as shown in Zhang et al. (2023) and Yang et al. (2024a), fully fine-tuning methods collapse to the empirical score instead of learning the target distribution and suffer from the memory phenomenon (only generate the target training dataset).

*Remark* 5.8 (Pretraining phase). Since the fully fine-tuning objective function is the same as the pretraining one (only different in the dataset), this result can also explain why the pretraining phase requires large optimization steps. More specifically, since the pretraining phase does not have the prior information of $\mu^*$, it is possible to initialize $\mu$ around 0, which leads to a small gradient.

## 6  GREAT PRETRAINING: WARM-UP AND ACCELERATION OPTIMIZATION

A significant advantage of the few-shot phase is that it can use the information of a well-trained score as the prior (such as latent information $\mu$ and data structure), which provides a warm-up for the few-shot phase. Based on this intuition, we show that few-shot models enjoy a simplified landscape and quickly converge to ground-truth parameters with a great pre-trained model.

To achieve this goal, we prove that the landscape of few-shot phase is strongly convex with great pretraining. As a start, we first show the form of $\partial^2 \mathcal{L}_{\mathrm{SM},t}^{\mathrm{few}}/\partial V_{ta}^2$, which consists two parts: the first squared term $N$ and the second cross term $M$ (we ignore $(x_t, t)$ and $\mathbb{E}_{x_t \sim q_t^{ta}}$ for clarity):

$$2\left(\frac{\partial s_{\hat\mu, V_{ta}}}{\partial V_{ta}}\right)^\top \left(\frac{\partial s_{\hat\mu, V_{ta}}}{\partial V_{ta}}\right) + 2\left(\frac{\partial^2 s_{\hat\mu, V_{ta}}}{\partial V_{ta}^2}\right)^\top (s_{\hat\mu, V_{ta}} - s_{\mu^*, A_{ta}}) := 2(N + M).$$

We know that the squared term $N$ is a semi-positive definite (SPD) matrix. However, due to the influence of the cross term, we determine a more precise lower bound for each element of $N$, as shown in the following lemma (In the following two lemmas, we ignore the $ta$ index of $v_{ta}$ and $V_{ta}$).

**Lemma 6.1.** *[Squared Term] Assume Assumption 3.1 and 5.1 holds and the latent parameter $\hat{\mu}$ is learning perfectly $\hat{\mu} = \mu^*$. $N \succeq \alpha I_2$ with $\alpha > 0$ for $\forall t \in [\delta, T]$ (see $\alpha$ in Eq.13).*

For the cross term, we provide an upper bound for each element to guarantee the negative influence is smaller than the positive influence of $N$.

**Lemma 6.2.** *[Cross Term] Following setting of Lem. 6.1. (a) **The** $|a_{ta} - v_{ta}| \leq \delta_{1,t}$ **situation.** For $\forall M(i,j), |M(i,j)| \leq \gamma(\delta_{1,t})$, where $\gamma(\delta_{1,t}) \to 0$ as $\delta_{1,t} \to 0$ (see $\gamma(\delta_{1,t})$ in Eq.15).*

*(b) **The** $v_{ta} \geq a_{ta} + \delta_{1,t}$ **situation.** Let $\delta_{2,t} \triangleq v_{ta} - a_{ta} \geq \delta_{1,t}$ and $M_1 = M - M'$, where $M'$ is SPD. Then, there exists an interval $v_{ta} \in [a_{ta} + \delta_{1,t}, a_{ta} + \delta_{2,t}]$ satisfies:*
$$\mathbb{E}[M_1(1,2)] = \mathbb{E}[M_1(2,1)] < 0, \mathbb{E}[M_1(1,1)] = \mathbb{E}[M_1(2,2)] > 0$$
$$\mathbb{E}[M_1(1,1) + M_1(1,2)] \geq u_1(v_{ta}, t) + u_2(v_{ta}, t),$$
*where $(u_1(v_{ta}, t) + u_2(v_{ta}, t))|_{v_{ta} = a_{ta} + \delta_{1,t}} > 0$, $u_1(\cdot, t)$ increasing and $u_2(\cdot, t)$ decreasing for $v_{ta} \in [a_{ta} + \delta_{1,t}, a_{ta} + \delta_{2,t}]$ (see $M', u_1(\cdot, t)$ and $u_2(\cdot, t)$ in Eq. 16, 17 and 18).*

Since the Hessian matrix $H = 2(M + N)$, if $\alpha \geq \gamma$, we know that $\mathcal{L}_{\mathrm{SM},t}^{\mathrm{few}}$ is $2(\alpha - \gamma)$-strongly convex for $|v_{ta} - a_{ta}| \leq \delta_{1,t}$. As shown in Lem. 6.2 (a), $\gamma$ is related to the initialization area, and we can determine a suitable initialization parameter $\delta_{1,t}$ to guarantee $\alpha \geq \gamma$. For the setting $v_{ta} \geq a_{ta} + \delta_{2,t}$, we only require $u_1(v_{ta}, t) + u_2(v_{ta}, t) \geq 0$. The following condition shows our requirement for initialization, and the example shows that the initialization requirement is easy to satisfy.

**Condition 1.** *$\delta_{1,t}$ satisfies $\alpha \geq \gamma(\delta_{1,t})$, and $\delta_{2,t}$ satisfies $u_1(a_{ta} + \delta_{2,t}) + u_2(a_{ta} + \delta_{2,t}) > 0$.*

**Example 6.3.** *Considering $A_s = [0.1, 0.1]^\top$, $A_{ta} = [0.12, 0.12]^\top$, and $\mu^* = 4$. With a $t = 2$, to satisfy **Condition 1**, we require $v_{ta}^{(0)} \in \{[0.1, 0.5] \cup [-0.5, -0.1]\}$, where $0.5$ is far away from $a_{ta}$.*

With similar idea, we upper bound the Hessian matrix and prove $\mathcal{L}_{\mathrm{SM},t}^{\mathrm{few}}$ is $2(\alpha + \gamma + \zeta)$-smooth with $\zeta \geq 0$ (see $\zeta$ in Eq. 19). Then, we have the following convergence guarantee for the few-shot process.

**Theorem 6.4.** *Assume Assumption 3.1, 5.1, $\hat{\mu} = \mu^*$ and $\delta_{1,t}, \delta_{2,t}$ satisfy **Condition 1**. Considering score matching function $\mathcal{L}_{\mathrm{SM},t}^{\mathrm{few}}$. When $v_{ta}^{(0)} \in \{[a_{ta} - \delta_{1,t}, a_{ta} + \delta_{2,t}] \cup [-a_{ta} - \delta_{2,t}, -a_{ta} + \delta_{1,t}]\}$, using gradient descent with learning rate $\eta = 1/(2\alpha + \zeta)$, with $\kappa = (\alpha + \gamma + \zeta)/(\alpha - \gamma)$, we have*

$$\left\| V_{ta}^{(k)} V_{ta}^{(k)\top} - A_{ta} A_{ta}^\top \right\|_F \leq \left( \frac{\kappa - 1}{\kappa + 1} \right)^k (2a_{ta} + \delta_{2,t}) |v_{ta}^{(0)} - a_{ta}|.$$

This result is the first convergence guarantee for the few-shot diffusion models and explains why few-shot models only require a few optimization steps to fast converge to the ground-truth parameter.

We also conduct simulation experiments to show the difference between the pretraining and few-shot phase and verify the landscape of $\mathcal{L}_{\mathrm{SM}}^{\mathrm{few}}$. More specifically, we calculate the Hessian with $\mu_t^*$ and different $v_{ta}$ and report the eigenvalues. Let $\lambda_1$ and $\lambda_2$ be the two smallest eigenvalues for $\mathcal{L}_{\mathrm{SM}}^{\mathrm{pre}}$ and $\lambda_1'$ and $\lambda_2'$ be the two for $\mathcal{L}_{\mathrm{SM}}^{\mathrm{few}}$. As shown in Table 2, the eigenvalue for the pretraining phase is negative, which indicates $\mathcal{L}_{\mathrm{SM}}^{\mathrm{pre}}$ is non-convex. On the contrary, the eigenvalues for the few-shot phase are positive. Hence, the few-shot objective function $\mathcal{L}_{\mathrm{SM}}^{\mathrm{few}}$ is strongly convex, which leads to a fast convergence rate and supports our Thm. 6.4. We provide similar simulation results under different $D, d$ (Appendix C), which indicates our theoretical results are representative and can be extended to general high-dimensional multi-modal GMM latent (Section 6.1).

| $v_{ta}$ | $\lambda_1$ | $\lambda_2$ | $\lambda_1'$ | $\lambda_2'$ |
|---|---|---|---|---|
| 0.07 | -2.8e-2 | -2.7e-2 | 2.5e-4 | 1.5e-3 |
| 0.2 | -7.1e-3 | -6.9e-3 | 0.033 | 0.034 |
| 0.3 | -2.6e-2 | -2.1e-2 | 0.0738 | 0.076 |
| 0.5 | -2.5e-2 | -1.5e-2 | 0.206 | 0.211 |

Table 2: $D = 8, d = 5$, 5-modal GMM latent.

*Remark* 6.5 (Influence of $t$.). Sec. 5 and 6 show that a great enough prior information $\hat{\mu}$ is important. When $t \to +\infty$, the information of $\hat{\mu}$ gradually disappears $\hat{\mu}_t \to 0$, which indicates the optimization process will become more difficult. Our convergence guarantee also reflects this intuition. When $t \to +\infty$, $\alpha$ (Eq. 13) and $\gamma$ (Eq. 15) become 0. As a result, the strongly convex parameter of the objective function becomes smaller, and the few-shot phase requires more optimization steps.

*Remark* 6.6 (Relationship with LoRA). We note that LoRA fine-tunes two low-rank matrices, and we also do this operation. Though our paradigm is slightly different from LoRA (LoRA fine-tunes two additional matrices), the optimization analysis for fine-tuning the linear matrices still makes a first step to explain the fast convergence process for a few-shot fine-tuning phase in the application.

## 6.1 DISCUSSION

In this part, we first discuss the estimation error introduced by the limited training datasets. Then, we provide the criteria of a great pretrained model from the theoretical and empirical perspectives. Finally, we intuitively show how to extend the analysis to a more general setting.

**Limited Source and Target Data.** For the pretraining phase, we assume the latent parameter is perfectly learned $\hat{\mu} = \mu^*$ in Sec. 6. In this part, we discuss the setting $\|\hat{\mu} - \mu^*\| \leq \epsilon_{\text{pre}}$ (given a pretraining dataset with $n_s$ datapoints, $\epsilon_{\text{pre}}$ has the order of $n_s^{-2/d}$ (Chen et al., 2023)). We know that when $\|\hat{\mu} - \mu^*\|$ increase, $\|\widehat{V}_{ta}\widehat{V}_{ta}^\top - A_{ta}A_{ta}^\top\|_F$ increase from 0 (Lem 5.3 and Thm. 5.4). Hence, when $\|\hat{\mu} - \mu^*\| \leq \epsilon_{\text{pre}}$ is small enough, $\|\widehat{V}_{ta}\widehat{V}_{ta}^\top - A_{ta}A_{ta}^\top\|_F \leq \text{poly}(\epsilon_{\text{pre}})$. For the few-shot phase, Yang et al. (2024a) show that there is an additional $1/\sqrt{n_{ta}}$ estimation error with $n_{ta}$ target data. Hence, there is an additional $\text{Poly}(n_s^{-2/d}) + n_{ta}^{-1/2}$ in Thm. 6.4 with a limited source and target data.

**Criteria of Great Pre-trained Model.** The performance of a pre-trained model is determined by the scale of pretrained data, the model size, and the optimization step. In Appendix B.1, we discuss the balance between these terms from the theoretical perspective. From the empirical perspective, a great pre-trained model is usually an over-parametrized NN (where large-scale models usually satisfy) and is trained with a large-scale and diverse dataset. The optimization process should choose a suitable step that enjoys a converged loss and has the ability to generalize. We can use the standard quality metric, such as FID, IS, Clip Score, Pickscore, etc., to avoid underfitting. For overfitting, one can sample multiple images with the same prompts and observe their diversity.

**Go Beyond: General Data.** This part intuitively discusses how to extend to general data. For high-dimensional multi-modal GMM latent, by calculating the Hessian w.r.t. $V$ with the closed-form score of $K$-modal GMM, we know that it still consists of the squared and cross terms, where the squared term is SPD, and the cross term tends to 0 as $s_\theta$ tends to $s^*$. Hence, intuitively, the square term can still overcome the cross term at a range around the ground truth score, which leads to a local convergence guarantee. As shown in Table 2, when the latent distribution is a general GMM distribution, the landscape of the few-shot phase is still strongly convex, which supports our intuition. To further support our discussion, we also provide a calculation and a rough bound in Appendix B.2. For the analysis of more general data distribution, we also discuss some methods and intuition to extend to more general bounded support latent (Appendix B.3) and multi low-dimensional linear subspace (Appendix B.4). We refer to Appendix B for more details.

## 7 CONCLUSION

This work aims to explain why few-shot diffusion models can achieve great performance with a few optimization steps. As a start, we first evaluate each operation of few-shot diffusion models and show that a bad pre-trained model heavily influences the few-shot phase through the real-world experiments. From the theoretical perspective, we prove that with a bad pre-trained model, the few-shot phase can not learn the ground truth parameters and suffers a small gradient, which highlights the importance of great pre-trained models. After that, we show that a great pretrained model provides a warm-up for the few-shot phase and makes the landscape of few-shot diffusion models strongly convex. As a result, we prove that the few-shot model can fast converge to the ground-truth parameters by using a standard optimization algorithm (such as gradient descent). Combined with the gradient analysis for the pretraining phase, for the first time, we explain why few-shot models only need a few optimization steps compared with the pretraining phase to achieve great performance.

**Future Work and Limitation.** In this work, we choose a 2-modal Gaussian mixture distribution as the latent distribution. Though this latent distribution is multi-modal and its score is nonlinear, there still exists a gap with the real-world data. In Appendix B, we discuss some promising extensions of few-shot analysis, including general GMM latent, general bounded support, and multiple low-dimensional manifolds. We left them as interesting future works.

**Ethics statement.** Our work aims to deepen the understanding of few-shot models and explain why few-shot diffusion models enjoy a fast convergence rate from a theoretical perspective. Since the few-shot diffusion only requires a limited amount of data and a few optimization steps, it can be used to generate deepfake images. To avoid this problem, we can add watermarking in the generated content Lu et al. (2024). Other societal impact is similar to general generative models (Mirsky and Lee, 2021).

**Reproducibility statement.** The detail and description of the real-world experiments are provided in Appendix C, including training and test datasets, neural network structure, the hyperparameters, and the training steps.

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

APPENDIX

# A   THE USE OF LARGE LANGUAGE MODELS (LLMS)

As this work mainly focus on theoretical guarantee for few-shot diffusion models, large language models were only used for grammar polishing. All ideas, real-world and simulation experiments, theoretical guarantee (estimation and optimization), discussion and writing decisions were conducted entirely by the authors without LLMs.

# B   MORE DISCUSSION ON GREAT PRE-TRAINED MODELS AND GENERAL LATENT

In this part, we first discuss the criteria of great pre-trained models and the influence of the pre-trained data, model size, and optimization steps.

## B.1   DISCUSSION ON THE GREAT PRE-TRAINED MODELS

As the performance (generalization or memorization) of pre-trained models is determined by the scale of data, the model size, and optimization steps, we discuss the different combinations of these components from a theoretical perspective, which is helpful in determining the criteria of great pre-trained models. After that, we discuss how to extend to a general GMM with latent and bounded support. Finally, we intuitively discuss how to extend the few-shot diffusion models analysis to general multi low-dimensional manifolds.

**The pretrained data.**   As discussed in Section 6.1, the limited pretrained data introduced the estimation error $\mathrm{Poly}(n_s^{-2/d})$ for the pretraining phase. From the theoretical perspective, we require that the imperfect learning error for the pretraining phase is not the dominant term, instead of perfect learning. More specifically, we require $\mathrm{Poly}(n_s^{-2/d})$ to be smaller than the optimization error in Thm. 6.4 and the limited target data error $n_{ta}^{-1/2}$.

**The balance between pretrained data, model size, and optimization step.**   The above discussion builds on the size of pre-trained data and NN matches (as shown in Theorem 2 of Chen et al. (2023), the size of NN is determined by $n_s$) and ignores the optimization steps. However, there exists a mismatch between the pretrained data, model size, and optimization step in the application. Based on Li et al. (2024), we discuss the influence of the following mismatch cases for the pre-trained models:

*Case 1: large train data and small NN size.*

In this setting, the NN tends to learn the Gaussian structure of pretrained data (the empirical mean and covariance) instead of learning the multi-modal information of data. This setting belongs to the underfitting bad pre-trained models since it can not greatly learn the knowledge of the source data.

*Case 2: Overparameterized NN with different optimization steps.*

When an NN is overparameterized, with a large enough optimization step, the NN will memorize the training data, leading to an overfitted, bad pre-trained model.

With a small optimization step, Zhang et al. (2023) show that the NN still learn the Gaussian structure of training data instead of total source data knowledge, which belongs to the underfitting bad pretrained model.

## B.2   THE ANALYSIS FOR THE GENERAL GMM LATENT.

Though the 2-mode GMM latent reflects the multi-modal property of real-world data, it still has some gap to the real-world complex data. In this part, we discuss how to extend our GMM latent to a general GMM latent. At the beginning, we calculate the Hessian and cross term. According to Shah et al. (2023), if the latent is a $K$-mode GMM, the score still has a closed form:

$$\nabla \log q_t(x) = A\Sigma_{i=1}^K \omega_{i,t}(A^\top x; \mu^*)\mu_{i,t}^* - AA^\top x - \frac{1}{\sigma_t^2}(I_D - AA^\top)X$$

where $e_{i,t}(x; \mu) = \exp(-||x - \mu_{i,t}||^2/2)$, $w_{i,t}(x; \mu) = \frac{e_{i,t}(x;\mu)}{\Sigma_{i=1}^K e_{i,t}(x;\mu)}$ and $\mu_{i,t} = \mu_i \exp(-t)$. When considering the $K$-mode GMM latent, we construct the NN with the above form and $s_{\hat{\mu}, V_{ta}}$ means use $\hat{\mu}$ and $V_{ta}$ to replace $\mu^*$ and $A_{ta}$ (For simplicity, we write $s_{\hat{\mu}, V_{ta}}$ with trainable $V_{ta}$ as $s_\theta$ and ignore the subscript $ta$). Noticing that

$$\frac{\partial^2 L}{\partial V^2} = 2 \underbrace{\left(\frac{\partial s_\theta}{\partial V}\right)\left(\frac{\partial s_\theta}{\partial V}\right)^\top}_{\text{Square term}} - 2 \underbrace{\frac{\partial^2 s_\theta}{\partial V^2}(s_\theta - s_{\mu^*, A_{ta}})}_{\text{Cross term}},$$

where the square term is a semi-positive definite matrix, and the cross term tends to 0 as $s_\theta$ tends to $s_{\mu^*, A_{ta}}$. So, our intuition is that

*The square term can overcome the cross term in the vicinity of the ground truth target when the distribution is a multivariate Gaussian mixture distribution.*

More specifically,

$$\frac{\partial s_\theta}{\partial V} = \underbrace{\Sigma_{i=1}^K \omega_{i,t}(V^\top x; \hat{\mu})\hat{\mu}_{i,t} I}_{P}$$

$$+ \underbrace{\Sigma_{i=1}^K \frac{\frac{\partial e_{i,t}}{\partial V}(\Sigma_{k=1}^K e_{k,t}) - \Sigma_{k=1}^K \frac{\partial e_{k,t}}{\partial V} e_{i,t}}{(\Sigma_{i=1}^K e_{k,t})^2} V^\top + \left(\frac{1}{\sigma_t^2} - 1\right)(xV^\top + V^\top X I)}_{Q}$$

We can know that $\lambda_{\min}\left(\left(\frac{\partial s_\theta}{\partial V}\right)^2\right) \geq (||P||_2 - ||Q||_2)^2 = c^2$, where $O(||P||_2) > O(||Q||_2)$ as $s_\theta$ tends to the ground truth target. This result means $\lambda_{\min}\left(\frac{\partial^2 L}{\partial V^2}\right) \approx c^2$ around the ground truth target, and the objective function is strongly convex (since the cross term can be ignored or only has a slight influence, which can be covered by $c^2$.

## B.3 EXTENSION TO GENERAL BOUNDED LATENT.

For the general latent distribution, if only focusing on the convergence guarantee, one possible way is to use the kernel-based method with a general wide 2-layer NN (the number of neurons $m = \Theta(n_s)$) (Li et al., 2023):

$$s_{t,A}(X) := A\text{ReLU}(WX + Ue(t))/m,$$

where $A \in \mathbb{R}^{D \times m}$ is trainable, $W \in \mathbb{R}^{m \times D}$ and $U \in \mathbb{R}^{m \times d_e}$ are randomly initialized and frozen during training, and $e(t)$ is the embedding of time. By setting $m = d$ (indicates $d$ is large enough, which is also used by Han et al. (2024a)), the trainable $A$ becomes the linear part, and the fixed $\text{ReLU}(WX + Ue(t))$ represents the nonlinear fixed latent in our work. Then, using the gradient flow algorithm, the score converges to the target distribution regardless of whether the pre-trained model is great (since the $W, U$ are randomly initialized). Though this method can provide a convergence guarantee, it does not reflect the role of pretrained models and does not match the empirical operation. Hence, we adopt a simple setting to clearly explain the optimization process of the few-shot phase.

## B.4 EXTENSION TO MULTI LOW-DIMENSIONAL SETTING

After obtaining the first convergence guarantee for the few-shot models under a single linear subspace with a GMM latent, we discuss how to extend the analysis to a union of low-dimensional subspaces.

Though real-world data admits the low-dimensional structure, it is a union of low-dimensional manifolds instead of one manifold (Brown et al., 2023). Hence, a setting closer to the real-world data is to assume the target data is a union of linear subspaces. In the pretraining phase, Wang et al. (2024) makes the first step in this direction by modeling the data as a union of linear subspaces, and each subspace admits a Gaussian latent. We can first follow this direction and extend it to the few-shot

phase. More specifically, for the few-shot modeling, we can assume the source and target data share some manifold and also have their own manifolds. Intuitively, since the pretraining phase has learn the shard manifold knowledge, based on our analysis, a great pre-trained model can also reduce the estimation error, warm-up, and accelerate the few-shot optimization process.

**Go beyond: Few-shot analysis for a union of linear manifolds with general GMM latent.** As Wang et al. (2024) assumes each manifold admits the Gaussian latent instead of the general GMM latent, it still has a gap to the real-world data. Another interesting future work is to combine the general GMM latent analysis (Appendix B.2) and multi-linear subspace assumption few-shot modeling to analyze the role of pre-trained models in the few-shot phase. We leave the analysis on the multi-subspace assumption and its GMM extension as interesting future works.

## C ADDITIONAL EXPERIMENTS

### C.1 ADDITIONAL SIMULATION EXPERIMENTS

In this part, we provide more simulation results with different $D$ and $d$ and show the two smallest eigenvalues of the Hessian matrix. As shown in the following two tables, the landscape of the pretraining phase is still non-convex. On the contrary, the landscape of the few-shot phase (with a great pretrained model) is almost strongly convex (except a very small negative eigenvalue $-7.5e-5$).

The non-convex landscape of the pretraining phase indicates that it is possible to converge to the local minima instead of the global minima. We also verify this intuition through the simulation experiment. More specifically, we use the initialization area $(v_{ta}, \mu_t^*)$ ($v_{ta} = 0.07$ and ground truth $a_{ta} = 0.12$) and update models with GD algorithm. Then, the pretraining phase converges to the local minima $0.112$, which is not equal to $a_{ta}$. On the contrary, the few-shot diffusion models with a fixed $\mu_t^*$ converge to $0.11999$, almost the same as $a_{ta}$.

| Value of $v_{ta}$ | $\lambda_1$ | $\lambda_2$ | $\lambda_1'$ | $\lambda_2'$ |
|---|---|---|---|---|
| 0.07 | -0.0013 | 0.0015 | 0.0007 | 0.0016 |
| 0.2 | -0.01 | 0.008 | 0.008 | 0.0083 |
| 0.3 | -0.027 | 0.012 | 0.0126 | 0.0133 |
| 0.5 | -0.057 | 0.013 | 0.0134 | 0.0151 |

Table 3: Eigenvalues for different $v_{ta}$ ($D = 16, d = 1$)

| Value of $v_{ta}$ | $\lambda_1$ | $\lambda_2$ | $\lambda_1'$ | $\lambda_2'$ |
|---|---|---|---|---|
| 0.07 | -0.0002 | -0.0002 | -7.5e-5 | 1.24e-7 |
| 0.2 | -0.0014 | -0.0008 | 1.18e-5 | 1.29e-5 |
| 0.3 | -0.0061 | -0.0047 | 2.67e-5 | 2.86e-5 |
| 0.5 | -0.0276 | -0.0264 | 7.42e-5 | 7.89e-5 |

Table 4: Eigenvalues for different $v_{ta}$ ($D = 4, d = 2$)

### C.2 THE DETAIL OF THE UNDERFITTING REAL-WORLD EXPERIMENTS

In this part, we describe the setting of our real-world experiments on the CelebA 64 datasets. Our setting mainly follows Yang et al. (2024a), and we provide the setting for the sake of completeness.

**CelabA64 Datasets.**

- Source dataset: 6400 images of faces with different hairstyles (without the bald feature).
- Target dataset: 10 images with the bald feature in CelebA64.

**NN structure.** In this experiment, we adopt a U-net network with 11 downblocks, 2 middleblocks, and 15 upblocks. In the pretraining phase, we train all parameters of the U-net. Since the NN layer in U-net is highly nonlinear, following Yang et al. (2024a), we fine-tune the downblock and upblocks in the few-shot fine-tuning phase. More specifically, we fine-tune the first 4 downblock layers (as the encoder) and 4 upblock layers (as the decoder) in the fine-tuning phase.

The above experiments were conducted on a GeForce RTX 4090. For the pre-trained phase, we train the models for 50 epochs (bad pretrained model, Model-50) with batch size 20, which takes 1 hour. The great pretrained model (Model-200) takes 5 hours in the pretraining phase. For the fine-tuning phase, we fine-tune the pre-trained models with limited target datasets for 400 epochs with a batch size of 2. It takes 3 minutes to fine-tune the pre-trained models.

### C.3 The Detail of the Overfitting Real-world Experiments

In the part, we provide the detail of the experiments on the Stable Diffusion models, including dataset and training pipeline.

**Dataset and Evaluation Metric.**
*Training Dataset.* The Dreambooth training dataset contains 30 subjects, and each subject contains 4-6 images to use to fine-tune the models (a total of 156 images).

*Validation Dataset.* The dreambooth dataset provides 25 test prompts for each subject (total $30 * 25 = 750$ prompts). Following the description of Dreambooth, we generate 4 images for each prompt and use these $3k$ images to evaluate.

*Clip-T Score.* Following Dreambooth, we calculate the cosine similarity between the prompt and image CLIP embeddings to measure the text-image alignment.

*Pickscore.* We also adopt the standard pickscore metric for text2image generation.

**Training Dreambooth pipeline.**
*Overfitting Bad Models.* Since the SD 1.4 and SD 3 Medium can generate high-quality and diverse samples, we view them as great pretrained models. To obtain a bad pretrained model, we overfit the SD 1.4 and SD 3 Medium with one prompt (a photo of a dog) and the corresponding 5 images. As the overfitting step increases, the diversity of pretrained models decreases (preferring to generate dog images in our setting). We use two bad pretrained models, the first one overfits the one prompt 1k steps, and the latter, a worse one, overfits 4k steps (lower diversity). The overfitting learning rate is $5 \times 10^{-6}$, the resolution is $512$ for SD 1.4 and $1024$ for SD3-Medium and the accumulation steps is $4$.

*Fine-tuning phase with freezing most parameters.* Then, we modify the train Dreambooth pipeline of the diffuser to train with the training dataset. To match the setting of our theoretical results, we only fine-tune the first and last 3 blocks of NN (Unet of SD 1.4 and DiT of SD 3). The fine-tuning optimization step is $1k$. The learning rate and resolution is the same with the overfitting phase.

## D The Detailed Calculation of Gradient and Hessian

Since our analysis depends heavily on the gradient and Hessian for the few-shot score matching objective function, we provide the detailed form of these terms in this section.

### D.1 Terms related to $\mathcal{L}_{\mathrm{SM},t}^{\mathrm{few}}$

Recall that $\mathbb{E}\frac{\partial^2 \mathcal{L}_{\mathrm{SM}}^{\mathrm{few}}}{\partial V_{ta}^2}$ is consisted by the cross and squared term

$$
\mathbb{E}\frac{\partial^2 \mathcal{L}_{\mathrm{SM}}^{\mathrm{few}}}{\partial V_{ta}^2} = 2\left[\mathbb{E}\left[\frac{\partial^2 s_{\hat{\mu},V_{ta}}}{\partial V_{ta}^2}\left(s_{\hat{\mu},V_{ta}} - s_{\mu^*,A_{ta}}\right)\right] + \mathbb{E}\left[\left(\frac{\partial s_{\hat{\mu},V_{ta}}}{\partial V_{ta}}\right)^{\top}\left(\frac{\partial s_{\hat{\mu},V_{ta}}}{\partial V_{ta}}\right)\right]\right].
$$

To obtain the exception form of $\mathbb{E}\frac{\partial^2 \mathcal{L}_{\mathrm{SM}}^{\mathrm{few}}}{\partial V_{ta}^2}$, we calculate the exception form of each term.

**Calculate $\frac{\partial \mathcal{L}_{\mathrm{SM},t}^{\mathrm{few}}}{\partial V_{ta}}$.** For this term, we know that

$$\frac{\partial \mathcal{L}_{\mathrm{SM},t}^{\mathrm{few}}}{\partial V_{ta}} = 2 \left( \frac{\partial s_{\hat{\mu},V_{ta}}}{\partial V_{ta}} \right)^{\top} (s_{\hat{\mu},V_{ta}}(x_t,t) - s_{\mu^*,A_{ta}}(x_t,t)) \,,$$

where

$$s_{\mu^*,A}(x_t,t) = A \tanh\left(\mu_t^{\star\top} A^{\top} x_t\right) \mu_t^* - AA^{\top} x_t - \frac{1}{\sigma_t^2} \left(I_D - AA^{\top}\right) x_t \,.$$

For the first term, we have the following equation:

$$\frac{\partial s_{\hat{\mu},V_{ta}}}{\partial V_{ta}} = \tanh(\hat{\mu}_t^{\top} V_{ta}^{\top} x_t)\hat{\mu}_t I_2 + \frac{\partial \tanh(\hat{\mu}_t^{\top} V_{ta}^{\top} x_t)\hat{\mu}_t}{\partial V_{ta}} V_{ta}^{\top} + \left(\frac{1}{\sigma_t^2} - 1\right) \frac{\partial V_{ta} V_{ta}^{\top} x_t}{\partial V_{ta}}$$

$$= \tanh(\hat{\mu}_t^{\top} V_{ta}^{\top} x_t)\hat{\mu}_t I_2 + (1 - \tanh^2(\hat{\mu}_t^{\top} V_{ta}^{\top} x_t))\hat{\mu}_t^{\top} \hat{\mu}_t x_t V_{ta}^{\top} + \left(\frac{1}{\sigma_t^2} - 1\right) (x_t V_{ta}^{\top} + V_{ta}^{\top} x_t I_2)$$

$$= \left(\tanh(\hat{\mu}_t^{\top} V_{ta}^{\top} x_t)\hat{\mu}_t + \left(\frac{1}{\sigma_t^2} - 1\right) V_{ta}^{\top} x_t\right) I_2 + \left((1 - \tanh^2(\hat{\mu}_t^{\top} V_{ta}^{\top} x_t))\hat{\mu}_t^{\top} \hat{\mu}_t + \left(\frac{1}{\sigma_t^2} - 1\right)\right) x_t V_{ta}^{\top} \,.$$

**Calculate $\frac{\partial^2 \mathcal{L}_{\mathrm{SM}}^{\mathrm{few}}}{\partial V_{ta}^2}$.** We know that the Hessian matrix of the few-shot score matching objective function can be decomposed into the cross term and the squared term.

$$\frac{\partial^2 \mathcal{L}_{\mathrm{SM}}^{\mathrm{few}}}{\partial V_{ta}^2} = 2 \underbrace{(\frac{\partial s_{\hat{\mu},V_{ta}}}{\partial V_{ta}})^{\top} \left(\frac{\partial s_{\hat{\mu},V_{ta}}}{\partial V_{ta}}\right)}_{\text{Squared Term } N} + 2 \underbrace{(\frac{\partial^2 s_{\hat{\mu},V_{ta}}}{\partial V_{ta}^2})^{\top} (s_{\hat{\mu},V_{ta}} - s_{\mu^*,A_{ta}})}_{\text{Cross Term } M} \,.$$

For the cross term, we know that

$$\frac{\partial^2 s_{\hat{\mu},V_{ta}}}{\partial V_{ta}^2}$$

$$= \begin{bmatrix} \left(1 - \tanh^2(\hat{\mu}_t^{\top} V_{ta}^{\top} x_t)\right) \hat{\mu}_t^{\top} \hat{\mu}_t \begin{bmatrix} x_t(1) & 0 \\ 0 & x_t(1) \end{bmatrix} \\ \left(1 - \tanh^2(\hat{\mu}_t^{\top} V_{ta}^{\top} x_t)\right) \hat{\mu}_t^{\top} \hat{\mu}_t \begin{bmatrix} x_t(2) & 0 \\ 0 & x_t(2) \end{bmatrix} \end{bmatrix} + 2\left(\frac{1}{\sigma_t^2} - 1\right) \begin{bmatrix} \begin{bmatrix} x_t(1) & 0 \\ 0 & x_t(1) \end{bmatrix} \\ \begin{bmatrix} x_t(2) & 0 \\ 0 & x_t(2) \end{bmatrix} \end{bmatrix}$$

$$+ \begin{bmatrix} \left(1 - \tanh^2(\hat{\mu}_t^{\top} V_{ta}^{\top} x_t)\right) \hat{\mu}_t^{\top} \hat{\mu}_t \begin{bmatrix} x_t(1) & 0 \\ x_t(2) & 0 \end{bmatrix} \\ \left(1 - \tanh^2(\hat{\mu}_t^{\top} V_{ta}^{\top} x_t)\right) \hat{\mu}_t^{\top} \hat{\mu}_t \begin{bmatrix} 0 & x_t(1) \\ 0 & x_t(2) \end{bmatrix} \end{bmatrix}$$

$$- 2\tanh(\hat{\mu}_t^{\top} V_{ta}^{\top} x_t)(1 - \tanh^2(\hat{\mu}_t^{\top} V_{ta}^{\top} x_t))\hat{\mu}_t^{\top} \hat{\mu}_t \begin{bmatrix} \hat{\mu}_t x_t(1) x_t V_{ta}^{\top} \\ \hat{\mu}_t x_t(2) x_t V_{ta}^{\top} \end{bmatrix} \,.$$

Let $s_{\hat{\mu},V_{ta}}(x_t,t) - s_{\mu^*,A_{ta}}(x_t,t) = y$. Then, we have that

$$(\frac{\partial^2 s_{\hat{\mu},V_{ta}}}{\partial V_{ta}^2})^{\top} (s_{\hat{\mu},V_{ta}} - s_{\mu^*,A_{ta}})$$

$$= (1 - \tanh^2(\hat{\mu}_t^{\top} V_{ta}^{\top} x_t))\hat{\mu}_t^{\top} \hat{\mu}_t x_t^{\top} y I_2 + (1 - \tanh^2(\hat{\mu}_t^{\top} V_{ta}^{\top} x_t))\hat{\mu}_t^{\top} \hat{\mu}_t x_t y^{\top}$$

$$- 2\tanh(\hat{\mu}_t^{\top} V_{ta}^{\top} x_t)(1 - \tanh^2(\hat{\mu}_t^{\top} V_{ta}^{\top} x_t))\hat{\mu}_t^{\top} \hat{\mu}_t \hat{\mu}_t x_t^{\top} y x_t V_{ta}^{\top} + 2\left(\frac{1}{\sigma_t^2} - 1\right) x_t^{\top} y I_2 \,.$$

For the squared term, we know that

$$\left(\frac{\partial s_{\hat{\mu}, V_{ta}}}{\partial V_{ta}}\right)^\top \left(\frac{\partial s_{\hat{\mu}, V_{ta}}}{\partial V_{ta}}\right) = \tanh^2(\hat{\mu}_t^\top V_{ta}^\top x_t)\hat{\mu}_t^\top \hat{\mu}_t I_2 + (1 - \tanh^2(\hat{\mu}_t^\top V_{ta}^\top x_t))^2 \hat{\mu}_t^\top \hat{\mu}_t V_{ta} x_t^\top x_t V_{ta}^\top$$

$$+ \left(\frac{1}{\sigma_t^2} - 1\right)^2 (V_{ta} x_t^\top x_t V_{ta}^\top + x_t^\top V_{ta} V_{ta}^\top x_t I_2 + V_{ta} x_t^\top V_{ta}^\top x_t + x_t^\top V_{ta} x_t V_{ta}^\top)$$

$$+ 2(1 - \tanh^2(\hat{\mu}_t^\top V_{ta}^\top x_t)) \tanh(\hat{\mu}_t^\top V_{ta}^\top x_t)\hat{\mu}_t \hat{\mu}_t^\top \hat{\mu}_t V_{ta} x_t^\top$$

$$+ 2\left(\frac{1}{\sigma_t^2} - 1\right) \tanh(\mu^\top V_{ta}^\top x_t)\hat{\mu}_t (x_t V_{ta}^\top + V_{ta}^\top x_t I_2)$$

$$+ 2(1 - \tanh^2(\hat{\mu}_t^\top V_{ta}^\top x_t)) \left(\frac{1}{\sigma_t^2} - 1\right) \hat{\mu}_t^\top \hat{\mu}_t V_{ta}^\top V_{ta} x_t x_t^\top$$

$$+ (1 - \tanh^2(\hat{\mu}_t^\top V_{ta}^\top x_t)) \left(\frac{1}{\sigma_t^2} - 1\right) \hat{\mu}_t^\top \hat{\mu}_t (x_t x_t^\top V_{ta} V_{ta}^\top + V_{ta} V_{ta}^\top x_t x_t^\top).$$

**Calculate the expectation of Hessian** $\mathbb{E}\frac{\partial^2 \mathcal{L}_{\mathrm{SM}}^{\mathrm{few}}}{\partial V_{ta}^2}$. As discussed in Section 6.1, we take expectation over the target distribution $q_{ta}$. Hence, we calculate the expectation of Hessian.

Before providing the result of the Hessian matrix, we first do some helpful calculation. Recall that under the linear subspace assumption (Assumption 3.1), the diffusion process happens in the latent distribution $z_0 \sim \frac{1}{2}\mathcal{N}(\mu^\star, 1) + \frac{1}{2}\mathcal{N}(-\mu^\star, 1)$, which indicates $z_t = \exp(-t)z_0 + \sqrt{1 - \exp(-2t)}\xi_t$ with $\xi_t \sim \mathcal{N}(0, 1)$. Then, by changing the probability density variable, we have

$$\exp(-t)z_0 = \frac{1}{2}\mathcal{N}(\exp(-t)\mu^\star, \exp(-2t)) + \frac{1}{2}\mathcal{N}(-\exp(-t)\mu^\star, \exp(-2t))$$

$$\sqrt{1 - \exp(-2t)}\xi_t \sim \mathcal{N}(0, (1 - \exp(-2t))$$

$$z_t = \exp(-t)z_0 + \sqrt{1 - \exp(-2t)}\xi_t \sim \frac{1}{2}\mathcal{N}(\mu_t^\star, 1) + \frac{1}{2}\mathcal{N}(-\mu_t^\star, 1).$$

Then, we know that $z_t \sim N(\mu_t^\star, 1)$, where $\mu_t^\star = \exp(-t)\mu^\star$, which indicates

$$x_t = A_{ta} z_t \sim \frac{1}{2}N(\mu_t^\star A_{ta}, A_{ta} A_{ta}^\top) + \frac{1}{2}N(-\mu_t^\star A_{ta}, A_{ta} A_{ta}^\top),$$

and

$$V_{ta}^\top x_t \sim \frac{1}{2}N(\mu_t^\star V_{ta}^\top A_{ta}, V_{ta}^\top A_{ta} A_{ta}^\top V_{ta}) + \frac{1}{2}N(-\mu_t^\star V_{ta}^\top A_{ta}, V_{ta}^\top A_{ta} A_{ta}^\top V_{ta}).$$

We should first calculate $\mathbb{E}[x_t x_t^\top]$, $\mathbb{E}[x_t^\top x_t]$, $\mathbb{E}[x_t^\top y]$ and $\mathbb{E}[x_t y^\top]$, where $y = s_{\hat{\mu}, V_{ta}} - s_{\mu^*, A_{ta}}$, as they will be frequently utilized in subsequent steps.

$$\mathbb{E}[x_t^\top x_t] = \mathbb{E}[\Sigma x_t(i)^2] = \Sigma D[x_t(i)] + \mathbb{E}^2[x_t(i)]$$
$$= tr(A_{ta} A_{ta}^\top) + \mathbb{E}[x_t]^\top \mathbb{E}[x_t]$$
$$= tr(A_{ta}^\top A_{ta}) + \mathbb{E}[x_t]^\top \mathbb{E}[x_t]$$
$$= (1 + \mu_t^{\star 2})A_{ta}^\top A_{ta}. \tag{4}$$

$$\mathbb{E}[x_t x_t^\top] = \mathbb{E}[(x_t - \mu^\star A_{ta})(x_t - \mu^\star A_{ta})^\top] + \mu^\star A_{ta}^\star \mathbb{E}[x_t^\top] + \mu^\star \mathbb{E}[x_t]A_{ta}^\top - \mu^{\star 2} A_{ta} A_{ta}^\top$$
$$= (1 + \mu^{\star 2})A_{ta} A_{ta}^\top$$

Observe that $x_t$ is a symmetric distribution. Then, for any even function $f$, we can write

$$\mathbb{E}_{x_t}[f(x_t)] = \frac{1}{2}\mathbb{E}_{x_t \sim \mathcal{N}(\mu_t^\star A_{ta}, A_{ta} A_{ta}^\top)}[f(x_t)] + \frac{1}{2}\mathbb{E}_{x_t \sim \mathcal{N}(-\mu_t^\star A_{ta}, A_{ta} A_{ta}^\top)}[f(x_t)]$$
$$= \mathbb{E}_{x_t \sim \mathcal{N}(\mu_t^\star A_{ta}, A_{ta} A_{ta}^\top)}[f(x_t)].$$

Applying this property of the even function, we can obtain the following result by using the fact that $x_t^\top y$ and $x_t y^\top$ are even functions $x_t$ (recall that $s_{\hat{\mu}, V_{ta}}(x_t, t) - s_{\mu^*, A_{ta}}(x_t, t) = y$).

$$\mathbb{E}[x_t^\top y]$$

$$= \mathbb{E}_{x_t}[x_t^\top (V_{ta} \tanh(\hat{\mu}_t^\top V_{ta}^\top x_t)\hat{\mu}_t - V_{ta}V_{ta}^\top x_t - \frac{1}{\sigma_t^2}(I_2 - V_{ta}V_{ta}^\top)x_t)] - \mathbb{E}_{x_t}[x_t^\top s_{\mu^*, A_{ta}}]$$

$$= \mathbb{E}_{x_t \sim N(\mu_t^\star A_{ta}, A_{ta}A_{ta}^\top)}[x_t^\top V_{ta}\tanh(\hat{\mu}_t^\top V_{ta}^\top x_t)\hat{\mu}_t + (\frac{1}{\sigma_t^2} - 1)x_t^\top V_{ta}V_{ta}^\top x_t - \frac{1}{\sigma_t^2}x_t^\top x_t)] \quad (5)$$

$$- \mathbb{E}_{x_t}[x_t^\top s_{\mu^*, A_{ta}}]$$

$$= \mathbb{E}_{x_t \sim N(\mu_t^\star A_{ta}, A_{ta}A_{ta}^\top)}[x_t^\top V_{ta}\tanh(\hat{\mu}_t^\top V_{ta}^\top x_t)\hat{\mu}_t] + (\frac{1}{\sigma_t^2} - 1)((1 + \mu_t^{\star 2})V_{ta}^\top A_{ta}A_{ta}^\top V_{ta})$$

$$- \frac{1}{\sigma_t^2}(tr(A_{ta}A_{ta}^\top) + \hat{\mu}_t^2 A_{ta}^\top A_{ta}) - \mathbb{E}_{x_t}[x_t^\top s_{\mu^*, A_{ta}}]$$

$$= \mathbb{E}_{x_t \sim N(\mu_t^\star A_{ta}, A_{ta}A_{ta}^\top)}[x_t^\top V_{ta}\tanh(\hat{\mu}_t^\top V_{ta}^\top x_t)\hat{\mu}_t] + (\frac{1}{\sigma_t^2} - 1)((1 + \mu_t^{\star 2})V_{ta}^\top A_{ta}A_{ta}^\top V_{ta})$$

$$- \frac{1}{\sigma_t^2}(tr(A_{ta}^\top A_{ta}) + \hat{\mu}_t^2 A_{ta}^\top A_{ta}) - \mathbb{E}_{x_t}[x_t^\top s_{\mu^*, A_{ta}}]$$

$$= \mathbb{E}_{x_t \sim N(\mu_t^* A_{ta}, A_{ta}A_{ta}^\top)}[x_t^\top V_{ta}\tanh(\hat{\mu}_t^\top V_{ta}^\top x_t)\hat{\mu}_t] + (\frac{1}{\sigma_t^2} - 1)((1 + \mu_t^{\star 2})V_{ta}^\top A_{ta}A_{ta}^\top V_{ta})$$

$$- \mathbb{E}_{x_t \sim N(\mu_t^\star A_{ta}, A_{ta}A_{ta}^\top)}[x_t^\top A_{ta}\tanh(\mu_t^{*\top} A_{ta}^\top x_t)\mu_t^*] - (\frac{1}{\sigma_t^2} - 1)((1 + \mu_t^{\star 2})A_{ta}^\top A_{ta}A_{ta}^\top A_{ta}) . \quad (6)$$

Through similar calculations, we can also get $\mathbb{E}_{x_t}[x_t y^\top]$:

$$\mathbb{E}_{x_t}[x_t y^\top] = \mathbb{E}[x_t(V_{ta}\tanh(\hat{\mu}_t^\top V_{ta}^\top x_t)\hat{\mu}_t - x_t^\top V_{ta}V_{ta}^\top - \frac{1}{\sigma_t^2}x_t^\top(I_2 - V_{ta}V_{ta}^\top)$$

$$- A_{ta}\tanh(\hat{\mu}_t^\top A_{ta}^\top x_t)\hat{\mu}_t - x_t^\top A_{ta}A_{ta}^\top - \frac{1}{\sigma_t^2}x_t^\top(I_2 - A_{ta}A_{ta}^\top))]$$

$$= \mathbb{E}_{x_t \sim N(\mu_t^\star A_{ta}, A_{ta}A_{ta}^\top)}\left[x_t(V_{ta}\tanh(\hat{\mu}_t^\top V_{ta}^\top x_t)\hat{\mu}_t - x_t(A_{ta}\tanh(\mu_t^{*\top} A_{ta}^\top x_t)\mu_t^*\right]$$

$$+ (1 + \mu_t^{\star 2})(\frac{1}{\sigma_t^2} - 1)A_{ta}A_{ta}^\top(V_{ta}V_{ta}^\top - A_{ta}A_{ta}^\top) . \quad (7)$$

With the calculation for $\mathbb{E}[x_t^\top y]$ and $\mathbb{E}_{x_t}[x_t y^\top]$, we can obtain the exception form of the cross and squared term. For the cross term, we have that

$$\mathbb{E}\left[\frac{\partial^2 s_{\hat{\mu}, V_{ta}}}{\partial V_{ta}^2}\left(s_{\hat{\mu}, V_{ta}} - s_{\mu^*, A_{ta}}\right)\right]$$

$$= \mathbb{E}\Big[(1 - \tanh^2(\hat{\mu}_t^\top V_{ta}^\top x_t))\hat{\mu}_t^\top \hat{\mu}_t x_t^\top y I_2 + (1 - \tanh^2(\hat{\mu}_t^\top V_{ta}^\top x_t))\hat{\mu}_t^\top \hat{\mu}_t x_t y^\top$$

$$- 2\tanh(\mu^\top V_{ta}^\top x_t)(1 - \tanh^2(\hat{\mu}_t^\top V_{ta}^\top x_t))\hat{\mu}_t^\top \hat{\mu}_t \hat{\mu}_t x_t^\top y x_t V_{ta}^\top + 2\left(\frac{1}{\sigma_t^2} - 1\right)x_t^\top y I_2\Big]$$

$$= \mathbb{E}\left[\hat{\mu}_t^\top \hat{\mu}_t x_t^\top y I_2\right] - \mathbb{E}\left[\tanh^2(\hat{\mu}_t^\top V_{ta}^\top x_t))\hat{\mu}_t^\top \hat{\mu}_t x_t^\top y\right] + \mathbb{E}\left[\hat{\mu}_t^\top \hat{\mu}_t x_t y^\top\right]$$

$$- \mathbb{E}\left[\tanh^2(\hat{\mu}_t^\top V_{ta}^\top x_t)\hat{\mu}_t^\top \hat{\mu}_t x_t y^\top\right]$$

$$- 2\mathbb{E}\left[\tanh(\hat{\mu}_t^\top V_{ta}^\top x_t)\hat{\mu}_t^\top \hat{\mu}_t \hat{\mu}_t x_t^\top y x_t V_{ta}^\top\right] + 2\mathbb{E}\left[\tanh^3(\hat{\mu}_t^\top V_{ta}^\top x_t)\hat{\mu}_t^\top \hat{\mu}_t \hat{\mu}_t x_t^\top y x_t V_{ta}^\top\right]$$

$$+ 2\left(\frac{1}{\sigma_t^2} - 1\right)\mathbb{E}[x_t^\top y I_2] .$$

For the squared term, we have that

$$
\mathbb{E}\left[\left(\frac{\partial s_{\hat{\mu}, V_{ta}}}{\partial V_{ta}}\right)^{\top}\left(\frac{\partial s_{\hat{\mu}, V_{ta}}}{\partial V_{ta}}\right)\right]
$$

$$
= \mathbb{E}[\tanh^2(\hat{\mu}_t^{\top} V_{ta}^{\top} x_t)\hat{\mu}_t^{\top}\hat{\mu}_t I_2] + \mathbb{E}[(1 - \tanh^2(\hat{\mu}_t^{\top} V_{ta}^{\top} x_t))^2 \hat{\mu}_t^{\top}\hat{\mu}_t V_{ta} x_t^{\top} x_t V_{ta}^{\top}]
$$

$$
+ 2\left(\frac{1}{\sigma_t^2} - 1\right)^2 ((1 + \hat{\mu}_t^2) A_{ta} A_{ta}^{\top} V_{ta} V_{ta}^{\top} + (1 + \hat{\mu}_t^2) V_{ta}^{\top} A_{ta} A_{ta}^{\top} V_{ta} I_2)
$$

$$
+ \mathbb{E}[2(1 - \tanh^2(\hat{\mu}_t^{\top} V_{ta}^{\top} x_t)) \tanh(\hat{\mu}_t^{\top} V_{ta}^{\top} x_t)\hat{\mu}_t^{\top}\hat{\mu}_t \hat{\mu}_t V_{ta} x_t^{\top}]
$$

$$
+ 2\mathbb{E}[(\frac{1}{\sigma_t^2} - 1) \tanh(\hat{\mu}_t^{\top} V_{ta}^{\top} x_t)\hat{\mu}_t(x_t V_{ta}^{\top} + V_{ta}^{\top} x_t I_2)]
$$

$$
+ \mathbb{E}(1 - \tanh^2(\hat{\mu}_t^{\top} V_{ta}^{\top} x_t))\left(\frac{1}{\sigma_t^2} - 1\right)\hat{\mu}_t^{\top}\hat{\mu}_t(2V_{ta}^{\top} V_{ta} x_t x_t^{\top} + x_t x_t^{\top} V_{ta} V_{ta}^{\top} + V_{ta} V_{ta}^{\top} x_t x_t^{\top}).
$$

## D.2 Terms related to $\mathcal{L}_{\mathrm{SM}}^{\mathrm{few-all}}$

For the fully fine-tuning method, we show that $\frac{\partial \mathcal{L}_{\mathrm{SM}}^{\mathrm{few-all}}}{\partial \mu_t}$ is small in Theorem 5.5. In this part, we provide the calculation of this term. We note that when considering fully fine-tuning method, $\mu_t$ also has a gradient.

$$
\frac{\partial \mathcal{L}_{\mathrm{SM}}^{\mathrm{few-all}}}{\partial \mu_t} = 2\left(s_{\mu, V_{ta}}(x_t, t) - s_{\mu^*, A_{ta}}\right)^{\top}\left(V_{ta}\tanh(\mu_t^{\top} V_{ta}^{\top} x_t) + \mu_t V_{ta}(1 - \tanh^2(\mu_t^{\top} V_{ta}^{\top} x_t))V_{ta}^{\top} x_t\right).
$$

# E The Proof for Bad Pretraining

**Lemma E.1.** *Assume Assumption 3.1 and 5.1 . If $\hat{\mu} \neq \mu^*$, with $V_{ta} V_{ta}^{\top} = A_{ta} A_{ta}^{\top}$, $\partial \mathcal{L}_{\mathrm{SM},t}^{\mathrm{few}} / \partial V_{ta} \neq 0$.*

**Proof.** For the sake of brevity, we use $x, \mu_t$ instead of $x_t^{ta}$, $\hat{\mu}_t$ when there is no ambiguity in this part. We also ignore $(x_t, t)$ in $s_{\hat{\mu}, V_{ta}}(x_t, t)$ and $s_{\mu^*, A_{ta}}(x_t, t)$ for clarity.

We know that

$$
\mathbb{E}\left[\frac{\partial \mathcal{L}_{\mathrm{SM},t}^{\mathrm{few}}}{\partial V_{ta}}\right] = \mathbb{E}\left[\left(\frac{\partial s_{\hat{\mu}, V_{ta}}}{\partial V_{ta}}\right)^{\top}(s_{\hat{\mu}, V_{ta}}(x, t) - s_{\mu^*, A_{ta}})(x, t)\right].
$$

For each term, we have the following form:

$$
\left(\frac{\partial s_{\hat{\mu}, V_{ta}}}{\partial V_{ta}}\right)^{\top} = \tanh(\mu_t V_{ta}^{\top} x)\mu_t I_2 + (1 - \tanh^2(\mu_t V_{ta}^{\top} x))\mu_t^{\top}\mu_t V_{ta} x^{\top}
$$

$$
+ \left(\frac{1}{\sigma_t^2} - 1\right)(V_{ta} x^{\top} + V_{ta}^{\top} x I_2)
$$

$$
\triangleq f(x, V_{ta}, \mu_t) + \left(\frac{1}{\sigma_t^2} - 1\right)(V_{ta} x^{\top} + V_{ta}^{\top} x I_2),
$$

and

$$
s_{\hat{\mu}, V_{ta}} - s_{\mu^*, A_{ta}} = V_{ta}\mu_t \tanh(\mu_t V_{ta}^{\top} x) - A_{ta}\mu_t^{\star} \tanh(\mu_t^{\star} A_{ta}^{\top} x) + \left(\frac{1}{\sigma_t^2} - 1\right)(V_{ta} V_{ta}^{\top} - A_{ta} A_{ta}^{\top})x
$$

$$
\triangleq g(x, V_{ta}, \mu_t, \mu_t^{\star}) + \left(\frac{1}{\sigma_t^2} - 1\right)(V_{ta} V_{ta}^{\top} - A_{ta} A_{ta}^{\top})x.
$$

Then, we simplify the gradient term into the following form

$$\mathbb{E}\left[\frac{\partial \mathcal{L}_{\mathrm{SM,t}}^{\mathrm{few}}}{\partial V_{ta}}\right] = \mathbb{E}\left[\left(\frac{\partial s_{\hat{\mu},V_{ta}}}{\partial V_{ta}}\right)^{\top}(s_{\hat{\mu},V_{ta}} - s_{\mu^*,A_{ta}})\right]$$

$$= \mathbb{E}\left[(f(x,V_{ta},\mu_t) + \left(\frac{1}{\sigma_t^2}-1\right)(V_{ta}x^{\top} + V_{ta}^{\top}xI_2))g(x,V_{ta},\mu_t,\mu_t^{\star})\right]$$

$$+ \mathbb{E}\left[(f(x,V_{ta},\mu_t) + \left(\frac{1}{\sigma_t^2}-1\right)(V_{ta}x^{\top} + V_{ta}^{\top}xI_2))\left(\frac{1}{\sigma_t^2}-1\right)(V_{ta}V_{ta}^{\top} - A_{ta}A_{ta}^{\top})x\right]$$

$$= \mathbb{E}_x\left[\left(\frac{1}{\sigma_t^2}-1\right)^2(V_{ta}x^{\top} + V_{ta}^{\top}xI_2)(V_{ta}V_{ta}^{\top} - A_{ta}A_{ta}^{\top})x] + \mathbb{E}_x[h(x,V_{ta},A_{ta},\mu_t,\mu_t^{\star})]\right],$$

where

$$h(x,V_{ta},A_{ta},\mu_t,\mu_t^{\star})$$

$$= f\left(\frac{1}{\sigma_t^2}-1\right)(V_{ta}V_{ta}^{\top} - A_{ta}A_{ta}^{\top})x + \left(\frac{1}{\sigma_t^2}-1\right)(V_{ta}x^{\top} + V_{ta}^{\top}xI_2)g + fg$$

$$= \left(\frac{1}{\sigma_t^2}-1\right)(\tanh(\mu_t V_{ta}^{\top}x)\mu_t I_2 + (1-\tanh^2(\mu_t V_{ta}^{\top}x))\mu_t^2 V_{ta}x^{\top})(V_{ta}V_{ta}^{\top} - A_{ta}A_{ta}^{\top})x$$

$$+ \left(\frac{1}{\sigma_t^2}-1\right)(V_{ta}x^{\top} + V_{ta}^{\top}xI_2)(V_{ta}\mu_t\tanh(\mu_t V_{ta}^{\top}x) - A_{ta}\mu_t^{\star}\tanh(\mu_t^{\star}A_{ta}^{\top}x))$$

$$+ (\tanh(\mu_t V_{ta}^{\top}x)\mu_t I_2$$

$$+ (1-\tanh^2(\mu_t V_{ta}^{\top}x))\mu_t^2 V_{ta}x^{\top})(V_{ta}\mu_t\tanh(\mu_t V_{ta}^{\top}x) - A_{ta}\mu_t^{\star}\tanh(\mu_t^{\star}A_{ta}^{\top}x)). \qquad (8)$$

We first calculate $\mathbb{E}_x[V_{ta}x^{\top}(V_{ta}V_{ta}^{\top} - A_{ta}A_{ta}^{\top})x]$ and $\mathbb{E}_x[(V_{ta}V_{ta}^{\top} - A_{ta}A_{ta}^{\top})xx^{\top}V_{ta}]$, which is useful in bounding the first term of $\mathbb{E}\left[\frac{\partial \mathcal{L}_{\mathrm{SM,t}}^{\mathrm{few}}}{\partial V_{ta}}\right]$:

$$\mathbb{E}_x[V_{ta}x^{\top}(V_{ta}V_{ta}^{\top} - A_{ta}A_{ta}^{\top})x] = V_{ta}\mathbb{E}_x[x^{\top}(V_{ta}V_{ta}^{\top} - A_{ta}A_{ta}^{\top})x]$$

$$= V_{ta}\mathbb{E}_x[tr(x^{\top}(V_{ta}V_{ta}^{\top} - A_{ta}A_{ta}^{\top})x)]$$

$$= V_{ta}\mathbb{E}_x[tr((V_{ta}V_{ta}^{\top} - A_{ta}A_{ta}^{\top})xx^{\top})]$$

$$= V_{ta}tr(\mathbb{E}_X[(V_{ta}V_{ta}^{\top} - A_{ta}A_{ta}^{\top})xx^{\top})]$$

$$= (1+\mu_t^{*2})tr((V_{ta}V_{ta}^{\top} - A_{ta}A_{ta}^{\top})A_{ta}A_{ta}^{\top})V_{ta},$$

where the last equality follows the fact that $\mathbb{E}[xx^{\top}] = (1+\mu_t^{*2})A_{ta}A_{ta}^{\top}$ (Eq.4). Similarly, we can obtain the following bound:

$$\mathbb{E}_x\left[(V_{ta}V_{ta}^{\top} - A_{ta}A_{ta}^{\top})xx^{\top}V_{ta}\right] = (V_{ta}V_{ta}^{\top} - A_{ta}A_{ta}^{\top})\mathbb{E}_X[xx^{\top}]V_{ta}$$

$$= (1+\mu_t^{*2})(V_{ta}V_{ta}^{\top} - A_{ta}A_{ta}^{\top})A_{ta}A_{ta}^{\top}V_{ta}.$$

Thus, the first term of the gradient $\mathbb{E}\left[\frac{\partial \mathcal{L}_{\mathrm{SM,t}}^{\mathrm{few}}}{\partial V_{ta}}\right]$ has the following form:

$$\mathbb{E}_x\left[\left(\frac{1}{\sigma_t^2}-1\right)^2(V_{ta}x^{\top} + V_{ta}^{\top}xI_2)(V_{ta}V_{ta}^{\top} - A_{ta}A_{ta}^{\top})x\right]$$

$$= \left(\frac{1}{\sigma_t^2}-1\right)^2\mathbb{E}_x[(V_{ta}x^{\top} + V_{ta}^{\top}xI_2)(V_{ta}V_{ta}^{\top} - A_{ta}A_{ta}^{\top})x]$$

$$= \left(\frac{1}{\sigma_t^2}-1\right)^2(\mathbb{E}_x[V_{ta}x^{\top}(V_{ta}V_{ta}^{\top} - A_{ta}A_{ta}^{\top})x] + \mathbb{E}[(V_{ta}V_{ta}^{\top} - A_{ta}A_{ta}^{\top})xx^{\top}V_{ta}])$$

$$= \left(\frac{1}{\sigma_t^2}-1\right)^2(1+\mu_t^{*2})((V_{ta}V_{ta}^{\top} - A_{ta}A_{ta}^{\top})A_{ta}A_{ta}^{\top}V_{ta} + tr((V_{ta}V_{ta}^{\top} - A_{ta}A_{ta}^{\top})A_{ta}A_{ta}^{\top})I_2)V_{ta}$$

$$\triangleq w(V_{ta}V_{ta}^{\top}, A_{ta}A_{ta})V_{ta}. \qquad (9)$$

Let $-\mathbb{E}_x[h(x, V_{ta}, A_{ta}, \mu_t, \mu_t^\star)] \triangleq h(V_{ta}, A_{ta}, \mu_t, \mu_t^\star)$, we know that

$$\mathbb{E}\left[\left(\frac{\partial s_{\hat{\mu}, V_{ta}}}{\partial V_{ta}}\right)^\top (s_{\hat{\mu}, V_{ta}} - s_{\mu^*, A_{ta}})\right] = 0$$

is equivalent to

$$w(V_{ta}V_{ta}^\top, A_{ta}A_{ta}^\top)V_{ta} = h(V_{ta}, A_{ta}, \mu_t, \mu_t^\star).$$

We then prove that $w(V_{ta}V_{ta}^\top, A_{ta}A_{ta}^\top) = 0$ if and only if $V_{ta}V_{ta}^\top = A_{ta}A_{ta}^\top$ when $A_{ta} \neq 0$.
If $V_{ta}V_{ta}^\top = A_{ta}A_{ta}^\top$:

$$V_{ta}V_{ta}^\top = A_{ta}A_{ta}^\top \Rightarrow V_{ta}V_{ta}^\top - A_{ta}A_{ta}^\top = 0$$
$$\Rightarrow w(V_{ta}V_{ta}^\top, A_{ta}A_{ta}^\top) = 0$$

If $w(V_{ta}V_{ta}^\top, A_{ta}A_{ta}^\top) = 0$, we know that

$$(V_{ta}V_{ta}^\top - A_{ta}A_{ta}^\top)A_{ta}A_{ta}^\top + tr((V_{ta}V_{ta}^\top - A_{ta}A_{ta}^\top)A_{ta}A_{ta}^\top)I_2 = 0$$
$$\Rightarrow tr((V_{ta}V_{ta}^\top - A_{ta}A_{ta}^\top)A_{ta}A_{ta}^\top) = -2tr((V_{ta}V_{ta}^\top - A_{ta}A_{ta}^\top)A_{ta}A_{ta}^\top),$$

which indicates $tr((V_{ta}V_{ta}^\top - A_{ta}A_{ta}^\top)A_{ta}A_{ta}^\top) = 0$. Then, we know that

$$V_{ta}V_{ta}^\top A_{ta}A_{ta}^\top - A_{ta}A_{ta}^\top A_{ta}A_{ta}^\top = -tr((AA^\top - A_{ta}A_{ta}^\top)A_{ta}A_{ta}^\top)I_2 = 0$$
$$\Rightarrow tr(V_{ta}V_{ta}^\top A_{ta}A_{ta}^\top) = V_{ta}^\top A_{ta}A_{ta}^\top V_{ta} = A_{ta}^\top A_{ta}A_{ta}^\top A_{ta}$$
$$\Rightarrow V_{ta}^\top A_{ta} = \pm A_{ta}^\top A_{ta}$$
$$\Rightarrow V_{ta}A_{ta}^\top A_{ta}A_{ta}^\top = \pm A_{ta}A_{ta}^\top A_{ta}A_{ta}^\top$$
$$\Rightarrow V_{ta}A_{ta}^\top = \pm A_{ta}A_{ta}^\top$$
$$\Rightarrow V_{ta} = \pm A_{ta}, V_{ta}V_{ta}^\top = A_{ta}A_{ta}^\top$$

Then we need $h(V_{ta}, A_{ta}, \mu_t, \mu_t^\star) = 0$. However, if $\mu_t \neq \mu_t^\star$, $h(V_{ta}, A_{ta}, \mu_t, \mu_t^\star) \neq 0$ when $V_{ta}V_{ta}^\top = A_{ta}A_{ta}^\top$. In other words, if $\mu_t \neq \mu_t^\star$, $V_{ta}V_{ta}^\top = A_{ta}A_{ta}^\top$ can not make $\mathbb{E}\left[\frac{\partial \mathcal{L}_{\text{SM,t}}^{\text{few}}}{\partial V_{ta}}\right] = 0$.
Then, we complete the proof. ∎

**Theorem 5.4.** *Assume Assumption 3.1 and 5.1 hold. Let $\mu_1^*$ and $\mu_2^*$ be the two parameters to generate different latent distributions. Given a bad pre-trained model with $\hat{\mu} = 0$, if $|\mu_1^* - \hat{\mu}| > |\mu_2^* - \hat{\mu}|$, then*

$$\|\widehat{V}_{ta,1}\widehat{V}_{ta,1}^\top - A_{ta}A_{ta}^\top\|_F > \|\widehat{V}_{ta,2}\widehat{V}_{ta,2}^\top - A_{ta}A_{ta}^\top\|_F,$$

*where $\widehat{V}_{ta,i}$ is the solution corresponds to $\mu_i^*, i \in \{1, 2\}$.*

**Proof.** With $\mu_t = 0$ and $\mu_t^\star \neq 0$, then we know that

$$h(V_{ta}, A_{ta}, \mu_t, \mu_t^\star) = \mathbb{E}_x\left[\left(\frac{1}{\sigma_t^2} - 1\right)(V_{ta}x^\top + V_{ta}^\top xI_2)\tanh(\mu_t^\star A_{ta}^\top x)\mu_t^\star\right]A_{ta}.$$

We know that

$$\mathbb{E}_x\left[\left(\frac{1}{\sigma_t^2} - 1\right)(x^\top A_{ta} + A_{ta}x^\top)\tanh(\mu_t^\star A_{ta}^\top x)\mu_t^\star\right] > 0$$

and

$$\mathbb{E}_x\left[\left(\frac{1}{\sigma_t^2} - 1\right)(x^\top A_{ta} + A_{ta}x^\top)\tanh(\mu_t^\star A_{ta}^\top x)\mu_t^\star\right] > 0,$$

so $w(V_{ta}V_{ta}^\top, A_{ta}A_{ta}^\top) > 0$, which means that there exists a constant positive gap between $V_{ta}V_{ta}^\top$ and $A_{ta}A_{ta}^\top$.

We also know that function $x \tanh x$ is even, which indicates if $\mu_1^\star > \mu_2^\star$,

$$\mathbb{E}_x \left[ \left( \frac{1}{\sigma_t^2} - 1 \right) (x^\top A_{ta} + A_{ta} x^\top) \tanh(\mu_1^\star A_{ta}^\top x) \mu_1^\star \right]$$

$$> \mathbb{E}_x \left[ \left( \frac{1}{\sigma_t^2} - 1 \right) (x^\top A_{ta} + A_{ta} x^\top) \tanh(\mu_2^\star A_{ta}^\top x) \mu_2^\star \right].$$

Therefore, the constant positive gap between $V_{ta} V_{ta}^\top$ and $A_{ta} A_{ta}^\top$ must increase.

$$\| \widehat{V}_{ta,1} \widehat{V}_{ta,1}^\top - A_{ta} A_{ta}^\top \|_F > \| \widehat{V}_{ta,2} \widehat{V}_{ta,2}^\top - A_{ta} A_{ta}^\top \|_F$$

∎

**Theorem 5.5.** *Assume Assumption 3.1 and 5.1 holds. For a fixed $t$, if $\mu_t \in (-\epsilon, \epsilon)$, we have that*

$$\partial \mathcal{L}_{\mathrm{SM},t}^{\mathrm{few-all}} / \partial \mu_t \le 4\epsilon A_{ta}^\top V_{ta} \sqrt{(1 + \mu_t^{\star 2}) V_{ta}^\top V_{ta}} \sqrt{C_1} + O(\epsilon^{\frac{3}{2}}),$$

*where $C_1$ is a small constant determined by $V_{ta}, A_{ta}$ and $\mu^*$ (Details in Eq. 10).*

**Proof.** Through simple algebraic calculations, we know the gradient for $\mu_t$ have the following form:

$$\frac{\partial \mathcal{L}_{\mathrm{SM}}^{\mathrm{few-all}}}{\partial \mu_t} = 2 \left( s_{\hat{\mu}, V_{ta}} - s_{\mu^*, A_{ta}} \right)^\top \left( V_{ta} \tanh(\mu_t^\top V_{ta}^\top x) + \mu_t V_{ta}(1 - \tanh(\mu_t^\top V_{ta}^\top x)) V_{ta}^\top x \right)$$

$$= 2 y^\top (V_{ta} \tanh(\mu_t^\top V_{ta}^\top x) + \mu_t V_{ta}(1 - \tanh(\mu_t^\top V_{ta}^\top x)) V_{ta}^\top x).$$

For the term, by using the Cauchy-Schwarz inequality, we know that

$$\mathbb{E}_x [2 y^\top (V_{ta} \tanh(\mu_t^\top V_{ta}^\top x) + \mu_t V_{ta}(1 - \tanh^2(\mu_t^\top V_{ta}^\top x)) V_{ta}^\top x)]$$

$$= \mathbb{E}_{x \sim \mathcal{N}(\mu^* A_{ta}, A_{ta} A_{ta}^\top)} [2 y^\top (V_{ta} \tanh(\mu_t^\top V_{ta}^\top x) + \mu_t V_{ta}(1 - \tanh^2(\mu_t^\top V_{ta}^\top x)) V_{ta}^\top x)]$$

$$\le 2 \sqrt{\mathbb{E}[y^\top y]} \times$$

$$\sqrt{\mathbb{E}[\| V_{ta} \tanh(\mu_t^\top V_{ta}^\top x) + \mu_t V_{ta}(1 - \tanh^2(\mu_t^\top V_{ta}^\top x)) V_{ta}^\top x) \|_2^2}.$$

Then we give the upper bounds on $\mathbb{E}[y^\top y]$ and

$$\mathbb{E} \left[ \| V_{ta} \tanh(\mu_t^\top V_{ta}^\top x) + \mu_t V_{ta}(1 - \tanh^2(\mu_t^\top V_{ta}^\top x)) V_{ta}^\top x) \|_2^2 \right]$$

to achieve the final bound.

For the second part, if $\mu_t \in (-\epsilon, \epsilon)$, we have

$$\left[ \| V_{ta} \tanh(\mu_t^\top V_{ta}^\top x) + \mu_t V_{ta}(1 - \tanh^2(\mu_t^\top V_{ta}^\top x)) V_{ta}^\top x) \|_2 \right]$$

$$\le \mathbb{E}[\epsilon^2 V_{ta}^\top V_{ta} V_{ta}^\top x x^\top V_{ta} + \epsilon^2 x^\top V_{ta} V_{ta}^\top V_{ta} V_{ta}^\top x + 2\epsilon^2 x^\top V_{ta} V_{ta}^\top V_{ta} V_{ta}^\top x]$$

$$= 4\epsilon^2 (1 + \mu_t^{\star 2}) V_{ta}^\top V_{ta} V_{ta}^\top A_{ta} A_{ta}^\top V_{ta},$$

where the inequality follows by the fact $\tanh^2(\mu_t^\top V_{ta}^\top x) \in [0, 1]$ and the first order of Taylor expansion for $\tanh(\mu_t^\top V_{ta}^\top x)$ (when $\mu_t \in (-\epsilon, \epsilon)$ is close to 0, the influence of higher-order terms in Taylor expansion is limited). The last equality follows Equation (4).

For the first part, we can divide $\mathbb{E}[y^\top y]$ into three parts below:

$$\mathbb{E}[y^\top y] = \mathbb{E}[\| V_{ta} \tanh(\mu_t V_{ta}^\top x) - A_{ta} \tanh(\mu_t^\star A_{ta}^\top x) \|_2^2]$$

$$+ 2\mathbb{E}[(\tanh(\mu_t V_{ta}^\top x) V_{ta}^\top - \tanh(\mu_t^\star A_{ta}^\top x) A_{ta}^\top)(V_{ta} V_{ta}^\top - A_{ta} A_{ta}^\top) x]$$

$$+ \left( \frac{1}{\sigma_t^2} - 1 \right)^2 tr((V_{ta} V_{ta}^\top - A_{ta} A_{ta}^\top)^2 A_{ta} A_{ta}^\top).$$

Next we bound each of these three terms separately.

**Bound for** $\mathbb{E}[\|V_{ta}\tanh(\mu_t V_{ta}^\top x) - A_{ta}\tanh(\mu_t^\star A_{ta}^\top x)\|_2^2]$.

$$\mathbb{E}\left[\|V_{ta}\tanh(\mu_t V_{ta}^\top x) - A_{ta}\tanh(\mu_t^\star A_{ta}^\top x)\|_2^2\right]$$

$$\leq \mathbb{E}\left[\epsilon^2 V_{ta}^\top V_{ta} V_{ta}^\top x x^\top V_{ta} + \mu_t^{\star 2} A_{ta}^\top A_{ta} A_{ta}^\top A_{ta} A_{ta}^\top x x^\top A_{ta} + 2\epsilon\mu_t^\star V_{ta}^\top A_{ta} V_{ta}^\top x x^\top A_{ta}\right]$$

$$= \left(\left(\epsilon^2 V_{ta}^\top V_{ta} V_{ta}^\top A_{ta} A_{ta}^\top V_{ta} + \mu_t^{\star 2} A_{ta}^\top A_{ta} A_{ta}^\top A_{ta} A_{ta}^\top A_{ta} A_{ta}^\top A_{ta} + 2\epsilon\mu_t^\star V_{ta}^\top A_{ta} A_{ta}^\top A_{ta} A_{ta}^\top A_{ta}\right)\right.$$

$$\left.\times (1 + \mu_t^{\star 2})\right) \triangleq C_1\,,$$

where the inequality follows by Equation (4) and the first order of Taylor expansion for $\tanh(\mu_t^\top V_{ta}^\top x)$ (when $\mu_t \in (-\epsilon, \epsilon)$ is close to 0, the influence of higher-order terms in Taylor expansion is limited).

**Bound for** $2\mathbb{E}[(\tanh(\mu_t V_{ta}^\top x)V_{ta}^\top - \tanh(\mu_t^\star A_{ta}^\top x)A_{ta}^\top)(V_{ta}V_{ta}^\top - A_{ta}A_{ta}^\top)x]$.

By simple algebraic calculation, we know that

$$2(\tanh(\mu_t V_{ta}^\top x)V_{ta}^\top - \tanh(\mu_t^\star A_{ta}^\top x)A_{ta}^\top)(V_{ta}V_{ta}^\top - A_{ta}A_{ta}^\top)x$$

$$= 2\Big(\tanh(\mu_t V_{ta}^\top x)V_{ta}^\top x V_{ta}^\top V_{ta} + \tanh(\mu_t^\star A_{ta}^\top x)A_{ta}^\top x A_{ta}^\top A_{ta}$$

$$\qquad\qquad - \tanh(\mu_t^\star A_{ta}^\top x)V_{ta}^\top x A_{ta}^\top V_{ta} - \tanh(\mu_t V_{ta}^\top x)A_{ta}^\top x A_{ta}^\top V_{ta}\Big)\,.$$

Then, we have the following bound

$$\mathbb{E}[(\tanh(\mu_t V_{ta}^\top x)V_{ta}^\top - \tanh(\mu_t^\star A_{ta}^\top x)A_{ta}^\top)(V_{ta}V_{ta}^\top - A_{ta}A_{ta}^\top)x]$$

$$\leq \Big(\epsilon(1 + \mu_t^{\star 2})V_{ta}^\top A_{ta}A_{ta}^\top V_{ta}V_{ta}^\top V_{ta} + \mu_t^\star(1 + \mu_t^{\star 2})A_{ta}^\top A_{ta}A_{ta}^\top A_{ta}A_{ta}^\top A_{ta}$$

$$\qquad\qquad + \epsilon(1 + \mu_t^{\star 2})V_{ta}^\top A_{ta}A_{ta}^\top A_{ta}A_{ta}^\top V_{ta}\Big) \triangleq C_2\,.$$

**Bound for** $\left(\frac{1}{\sigma_t^2} - 1\right)^2 tr((V_{ta}V_{ta}^\top - A_{ta}A_{ta}^\top)^2 A_{ta}A_{ta}^\top)$.

$$\left(\frac{1}{\sigma_t^2} - 1\right)^2 tr((V_{ta}V_{ta}^\top - A_{ta}A_{ta}^\top)^2 A_{ta}A_{ta}^\top)$$

$$= \left(\frac{1}{\sigma_t^2} - 1\right)^2 [V_{ta}^\top V_{ta}(V_{ta}^\top A_{ta})^2 - 2A_{ta}^\top A_{ta}(V_{ta}^\top A_{ta})^2 + (A_{ta}^\top A_{ta})^3] \triangleq C_3\,.$$

Then, we know that

$$\mathbb{E}[y^\top y] \leq C_1 + 2C_2 + C_3 \triangleq C\,.$$

For $\forall V_{ta}$, $C = C' + O(\epsilon)$, while

$$C' = (1 + \mu_t^{\star 2})\mu_t^{\star 2}(A_{ta}^\top A_{ta})^3 + 2\mu_t^\star(1 + \mu_t^{\star 2})(A_{ta}^\top A_{ta})^3 + C_3$$

$$= 3(1 + \mu_t^{\star 2})\mu_t^{\star 2}(A_{ta}^\top A_{ta})^3 + C_3 < +\infty\,. \tag{10}$$

Then, we obtain the following bound for the gradient of fully fine-tuning method:

$$\mathbb{E}_x\left[\frac{\partial \mathcal{L}_{\text{SM}}^{\text{few}-\text{all}}}{\partial \mu_t}\right] \leq 4\epsilon A_{ta}^\top V_{ta}\sqrt{(1 + \mu_t^{\star 2})V_{ta}^\top V_{ta}}\sqrt{C'} + O(\epsilon^{\frac{3}{2}})$$

When $\epsilon \leq \frac{1}{4V_{ta}^\top A_{ta}\sqrt{C}\sqrt{(1+\mu_t^{\star 2})V_{ta}^\top V_{ta}}} \times 10^{-5}$, $\mathbb{E}_x[\frac{\partial \mathcal{L}}{\partial \mu_t}] \leq 1 \times 10^{-5}$, which indicates that large optimization steps are required in the optimization process.

Under the setting of Example 11, if $V_{ta} = [0.1, 0.1]^\top$, $\mathbb{E}_x\left[\frac{\partial \mathcal{L}_{\text{SM},t}^{\text{few}-\text{all}}}{\partial \mu_t}\right] \leq 1 \times 10^{-5}$ when $\epsilon < 0.12$. $\blacksquare$

## F   THE PROOF FOR GOOD PRETRAINING

In this section, by analyzing the Hessian of score matching objective function for the few-shot phase $\frac{\partial^2 \mathcal{L}_{\text{SM},t}^{\text{few}}}{\partial V_{ta}^2}$, we prove that with a large initialization area, the objective function is strongly convex, which leads to a convergence guarantee.

Since we assume the latent parameter $\mu^*$ is perfectly learned by the pretraining phase $\hat{\mu} = \mu^*$, we do not especially distinguish $\hat{\mu}_t, \mu_t^*$ and $\mu_t$ in the proof of this section. We also ignore the subscript $t$ of $x$ when there is no ambiguity. Furthermore, since some results rely on the initialization area, we use the following simple example to show how to satisfy the requirement after providing the theoretical guarantee.

**Example Setting**

$$t = 2, \mu^* = 4, A_s = [[0.1, 0.1]]^\top \text{ and } A_{ta} = [[0.12, 0.12]]^\top. \tag{11}$$

Recall that the Hessian has the following form

$$2 \underbrace{\left(\frac{\partial s_{\hat{\mu}, V_{ta}}}{\partial V_{ta}}\right)^\top \left(\frac{\partial s_{\hat{\mu}, V_{ta}}}{\partial V_{ta}}\right)}_{\text{Squared Term } N} + 2 \underbrace{\left(\frac{\partial^2 s_{\hat{\mu}, V_{ta}}}{\partial V_{ta}^2}\right)^\top (s_{\hat{\mu}, V_{ta}} - s_{\mu^*, A_{ta}})}_{\text{Cross Term } M}.$$

First we analyze term $MM^\top$, where $M$ has the form as $aI + bxy^\top$, which will be used in the following lemma.

**Lemma F.1.** *Let $M = aI + bxy^\top$, $MM^\top$ is semi-positive definite. And it's positive definite if and only if $a = 0$ or $a + bx^\top y = 0$.*

$$\lambda_{\min}(MM^\top)$$
$$= \min\left(a^2, a^2 + abx^\top y + \frac{b^2 \|x\|^2 \|y\|^2}{2} - \frac{b}{2}\sqrt{\|x\|^2 \|y\|^2 \left(4a^2 + 4abx^\top y + b^2 \|x\|^2 \|y\|^2\right)}\right)$$

**Proof.** First, $\forall x \in \mathbb{R}^d$, we have

$$x^\top MM^\top x = (M^\top x)^\top (M^\top x)$$
$$= \|M^\top x\|_2 \geq 0$$

Thus, $MM^\top$ is semi-positive definite.

We can also obtain that

$$|aI + bxy^\top| = |aI|(1 + \frac{b}{a}x^\top y)$$
$$= a^{n-1}(a + bx^\top y)$$

Therefore,

$$|MM^\top| = a^{2n-2}(a + bx^\top y)^2 \geq 0,$$

the equality holds if and only if $a = 0$ or $a + bx^\top y = 0$.

We further derive the eigenvalues of $MM^\top$.

Let $\lambda$ be an eigenvalue of $M^\top M$ with corresponding eigenvector $v$.

$$(M^\top M)v = \lambda v$$

We can analyze the action of this matrix on two orthogonal subspaces.

Let $S = \text{span}\{x, y\}$. Consider a vector $v$ in the orthogonal complement of $S$, denoted $S^\perp$. For any such vector $v \neq \mathbf{0}$, we have $x^\top v = 0$ and $y^\top v = 0$.

Let's apply $M^\top M$ to $v$:

$$(M^\top M)v = (a^2 I + ab(xy^\top + yx^\top) + b^2\|x\|^2 yy^\top)v$$
$$= a^2 Iv + ab(x(y^\top v) + y(x^\top v)) + b^2\|x\|^2 y(y^\top v)$$
$$= a^2 v + ab(x(0) + y(0)) + b^2\|x\|^2 y(0)$$
$$= a^2 v$$

This shows that any vector $v$ orthogonal to both $x$ and $y$ is an eigenvector of $M^\top M$ with the eigenvalue $\lambda = a^2$. The dimension of this subspace, $\dim(S^\perp)$, is at least $n-2$. Therefore, $a^2$ is an eigenvalue of $M^\top M$ with a multiplicity of at least $n-2$.

For the other 2 eigenvalues, we set them $\mu_1$ and $\mu_2$. We know the determinant of a matrix is the product of its eigenvalues.

$$\det(M^\top M) = (a^2)^{n-2}\mu_1\mu_2$$

We also know that $\det(M^\top M) = \det(M^\top)\det(M) = (\det(M))^2$. The determinant of the original matrix $M$ is $\det(M) = a^{n-1}(a + by^\top x)$. Therefore:

$$\det(M^\top M) = \left[a^{n-1}(a + by^\top x)\right]^2 = a^{2n-2}(a + by^\top x)^2$$

Equating the two expressions for the determinant:

$$a^{2n-4}\mu_1\mu_2 = a^{2n-2}(a + by^\top x)^2$$

Solving for the product $\mu_1\mu_2$ (assuming $a \neq 0$):

$$\mu_1\mu_2 = a^2(a + by^\top x)^2$$

The trace of a matrix is the sum of its eigenvalues.

$$\mathrm{tr}(M^\top M) = (n-2)a^2 + \mu_1 + \mu_2$$

We can also compute the trace directly from the expression for $M^\top M$:

$$\mathrm{tr}(M^\top M) = \mathrm{tr}(a^2 I + ab(xy^\top + yx^\top) + b^2\|x\|^2 yy^\top)$$

Using the linearity of the trace and the property $\mathrm{tr}(AB) = \mathrm{tr}(BA)$:

$$\mathrm{tr}(M^\top M) = a^2\mathrm{tr}(I) + ab(\mathrm{tr}(xy^\top) + \mathrm{tr}(yx^\top)) + b^2\|x\|^2\mathrm{tr}(yy^\top)$$
$$= na^2 + ab(y^\top x + x^\top y) + b^2\|x\|^2(y^\top y)$$
$$= na^2 + 2ab(y^\top x) + b^2\|x\|^2\|y\|^2$$

Equating the two expressions for the trace:

$$(n-2)a^2 + \mu_1 + \mu_2 = na^2 + 2ab(y^\top) + b^2\|x\|^2\|y\|^2$$

Solving for the sum $\mu_1 + \mu_2$:

$$\mu_1 + \mu_2 = 2a^2 + 2ab(y^\top x) + b^2\|x\|^2\|y\|^2$$

Thus, $\mu_1$ and $\mu_2$ are two solutions of

$$\mu^2 - \left(2a^2 + 2ab(y^\top x) + b^2\|x\|^2\|y\|^2\right)\mu + a^2(a + by^\top x)^2 = 0$$

We finally obtain that

$$\lambda_{\min}(MM^\top)$$
$$= \min\left(a^2,\ a^2 + abx^\top y + \frac{b^2\|x\|^2\|y\|^2}{2} - \frac{b}{2}\sqrt{\|x\|^2\|y\|^2\left(4a^2 + 4abx^\top y + b^2\|x\|^2\|y\|^2\right)}\right)$$

∎

In the following two lemmas, we provide the bound for the squared term and cross term, respectively.

**Lemma F.2.** *[Squared Term] Assume Assumption 3.1 and 5.1 holds and the latent parameter $\hat{\mu}$ is learning perfectly $\hat{\mu} = \mu^*$. $N \succeq \alpha I_2$ with $\alpha > 0$ for $\forall t \in [\delta, T]$ (see $\alpha$ in Eq.13).*

**Proof.** Recall that

$$\frac{\partial s_{\hat{\mu}, V_{ta}}}{\partial V_{ta}} = \tanh(\hat{\mu}_t^\top V_{ta}^\top x_t)\hat{\mu}_t I_2 + \frac{\partial \tanh(\hat{\mu}_t^\top V_{ta}^\top x_t)\hat{\mu}_t}{\partial V_{ta}} V_{ta}^\top + \left(\frac{1}{\sigma_t^2} - 1\right) \frac{\partial V_{ta} V_{ta}^\top x_t}{\partial V_{ta}}$$

$$= \tanh(\hat{\mu}_t^\top V_{ta}^\top x_t)\hat{\mu}_t I_2 + (1 - \tanh^2(\hat{\mu}_t^\top V_{ta}^\top x_t))\hat{\mu}_t^\top \hat{\mu}_t x_t V_{ta}^\top + \left(\frac{1}{\sigma_t^2} - 1\right)(x_t V_{ta}^\top + V_{ta}^\top x_t I_2)$$

$$= \left(\tanh(\hat{\mu}_t^\top V_{ta}^\top x_t)\hat{\mu}_t + \left(\frac{1}{\sigma_t^2} - 1\right) V_{ta}^\top x_t\right) I_2$$

$$+ \left((1 - \tanh^2(\hat{\mu}_t^\top V_{ta}^\top x_t))\hat{\mu}_t^\top \hat{\mu}_t + \left(\frac{1}{\sigma_t^2} - 1\right)\right) x_t V_{ta}^\top.$$

Let $p = \tanh(\hat{\mu}_t^\top V_{ta}^\top x_t)\hat{\mu}_t + \left(\frac{1}{\sigma_t^2} - 1\right) V_{ta}^\top x_t$ and $q = (1 - \tanh^2(\hat{\mu}_t^\top V_{ta}^\top x_t))\hat{\mu}_t^\top \hat{\mu}_t + \frac{1}{\sigma_t^2} - 1$, the squared term can be simplified as:

$$\mathbb{E}_x\left[\left(\frac{\partial s_{\hat{\mu}, V_{ta}}}{\partial V_{ta}}\right)^\top \left(\frac{\partial s_{\hat{\mu}, V_{ta}}}{\partial V_{ta}}\right)\right] = \mathbb{E}_x\left[(pI_2 + qx_t V_{ta}^\top)(pI_2 + qx_t V_{ta}^\top)^\top\right].$$

Using lemma F.1, we can obtain that

$$\lambda_{min}((pI_2 + qx_t V_{ta}^\top)(pI_2 + qx_t V_{ta}^\top)^\top)$$

$$= \min\left(p^2, \; p^2 + pqx^\top V_{ta} + \frac{q^2\|x\|^2\|V_{ta}\|^2}{2} - \frac{q}{2}\sqrt{\|x\|^2\|V_{ta}\|^2 (4p^2 + 4pqx^\top V_{ta} + q^2\|x\|^2\|V_{ta}\|^2)}\right),$$

where $p = \tanh(\hat{\mu}_t^\top V_{ta}^\top x_t)\hat{\mu}_t + \left(\frac{1}{\sigma_t^2} - 1\right) V_{ta}^\top x_t$ and $q = (1 - \tanh^2(\hat{\mu}_t^\top V_{ta}^\top x_t))\hat{\mu}_t^\top \hat{\mu}_t + \frac{1}{\sigma_t^2} - 1 > 0$. Moreover, since $q > 0$, we have

$$2pV_{ta}^\top x_t + q\|x\|^2\|V_{ta}\|^2 \le \|x\|\|V_{ta}\|\sqrt{4p^2 + 4pqV_{ta}^\top x_t + q^2\|x\|^2\|V_{ta}\|^2}, \tag{12}$$

the equality holds if and only if $x_t = kV_{ta}$.

The inequality 12 holds because of the Cauchy-Schwarz Inequality, which can be used through squaring both sides and rearranging the terms.

Thus,

$$pqx^\top V_{ta} + \frac{q^2\|x\|^2\|V_{ta}\|^2}{2} - \frac{q}{2}\sqrt{\|x\|^2\|V_{ta}\|^2 (4p^2 + 4pqx^\top V_{ta} + q^2\|x\|^2\|V_{ta}\|^2)}$$

$$= \frac{q}{2}(2pV_{ta}^\top x_t + q\|x\|^2\|V_{ta}\|^2 - \|x\|\|V_{ta}\|\sqrt{4p^2 + 4pqV_{ta}^\top x_t + q^2\|x\|^2\|V_{ta}\|^2}) \le 0,$$

and

$$\lambda_{min}((pI_2 + qx_t V_{ta}^\top)(pI_2 + qx_t V_{ta}^\top)^\top)$$

$$= p^2 + pqx^\top V_{ta} + \frac{q^2\|x\|^2\|V_{ta}\|^2}{2} - \frac{q}{2}\sqrt{\|x\|^2\|V_{ta}\|^2 (4p^2 + 4pqx^\top V_{ta} + q^2\|x\|^2\|V_{ta}\|^2)}.$$

After analyzing each term, we can choose $N_1 = \alpha I_2$ with

$$\alpha \triangleq \mathbb{E}_{x \sim p_{data}}\left[p^2 + pqx^\top V_{ta} + \frac{q^2\|x\|^2\|V_{ta}\|^2}{2} - \frac{q}{2}\sqrt{\|x\|^2\|V_{ta}\|^2 (4p^2 + 4pqx^\top V_{ta} + q^2\|x\|^2\|V_{ta}\|^2)}\right], \tag{13}$$

where $p = \tanh(\hat{\mu}_t^\top V_{ta}^\top x_t)\hat{\mu}_t + \left(\frac{1}{\sigma_t^2} - 1\right) V_{ta}^\top x_t$ and $q = (1 - \tanh^2(\hat{\mu}_t^\top V_{ta}^\top x_t))\hat{\mu}_t^\top \hat{\mu}_t + \frac{1}{\sigma_t^2} - 1 > 0$.

Then, we complete our proof.

$\blacksquare$

For the cross term, we analyze two situations: the initialization area is around the ground-truth $a_{ta}$: $|a_{ta} - v_{ta}| \leq \delta_{1,t}$ and initialization area is on the right hand of $a_{ta}$: $v_{ta} \geq a_{ta} + \delta_{1,t}$. When $v_{ta} \leq a_{ta}$, it is possible for the cross term $M$ to be the negative definite matrix. Hence, we control each element to guarantee the negative influence of the negative definite matrix is small. When $v_{ta} \geq a_{ta}$, the cross term $M$ is semi-positive definite in a large region.

**Lemma F.3.** *[Cross Term] Following setting of Lem. 6.1. (a) **The** $|a_{ta} - v_{ta}| \leq \delta_{1,t}$ **situation.** For $\forall M(i,j)$, $|M(i,j)| \leq \gamma(\delta_{1,t})$, where $\gamma(\delta_{1,t}) \to 0$ as $\delta_{1,t} \to 0$ (see $\gamma(\delta_{1,t})$ in Eq.15).*

*(b) **The** $v_{ta} \geq a_{ta} + \delta_{1,t}$ **situation.** Let $\delta_{2,t} \triangleq v_{ta} - a_{ta} \geq \delta_{1,t}$ and $M_1 = M - M'$, where $M'$ is SPD. Then, there exists an interval $v_{ta} \in [a_{ta} + \delta_{1,t}, a_{ta} + \delta_{2,t}]$ satisfies:*

$$\mathbb{E}[M_1(1,2)] = \mathbb{E}[M_1(2,1)] < 0, \mathbb{E}[M_1(1,1)] = \mathbb{E}[M_1(2,2)] > 0$$
$$\mathbb{E}[M_1(1,1) + M_1(1,2)] \geq u_1(v_{ta}, t) + u_2(v_{ta}, t),$$

*where $(u_1(v_{ta}, t) + u_2(v_{ta}, t))|_{v_{ta}=a_{ta}+\delta_{1,t}} > 0$, $u_1(\cdot, t)$ increasing and $u_2(\cdot, t)$ decreasing for $v_{ta} \in [a_{ta} + \delta_{1,t}, a_{ta} + \delta_{2,t}]$ (see $M'$, $u_1(\cdot, t)$ and $u_2(\cdot, t)$ in Eq. 16, 17 and 18).*

**Proof.** We know that the cross term has the following form (in this lemma, we ignore the subscript $t$ of $x$.)

$$\mathbb{E}\left[ \frac{\partial^2 s_{\hat{\mu}, V_{ta}}}{\partial V_{ta}^2} (s_{\hat{\mu}, V_{ta}} - s_{\mu^*, A_{ta}}) \right]$$
$$= \mathbb{E}[(1 - \tanh^2(\mu_t^\top V_{ta}^\top x))\mu_t^\top \mu_t x^\top y I_2 + (1 - \tanh^2(\mu_t^\top V_{ta}^\top x))\mu_t^\top \mu_t x y^\top$$
$$- 2\tanh(\mu_t^\top V_{ta}^\top x)(1 - \tanh^2(\mu_t^\top V_{ta}^\top x))\mu_t^\top \mu_t \mu_t x^\top y x V_{ta}^\top + 2\left(\frac{1}{\sigma_t^2} - 1\right) x^\top y I_2]. \quad (14)$$

We want to make

$$\mathbb{E}\left[ \frac{\partial^2 s_{\hat{\mu}, V_{ta}}}{\partial V_{ta}^2} (s_{\hat{\mu}, V_{ta}} - s_{\mu^*, A_{ta}}) \right] + 2(\frac{1}{\sigma_t^2} - 1)^2(1 + \mu_t^2)A_{ta}^\top A_{ta} V_{ta}^\top V_{ta} I_2$$
$$+ \mu_t^2(\frac{1}{\sigma_t^2} - 1)\tanh(\mu_t^2 V_{ta}^\top A_{ta})V_{ta}^\top A_{ta} I_2 + \mathbb{E}_X[\tanh^2(\mu_t^\top V_{ta}^\top x)\mu_t^2]I_2$$

positive definite, where the last three terms come from the above squared term.

In the proof of this lemma, we redefine $x$:

$$x = [x(1), x(2)]^\top \sim \mathcal{N}(\mu_t A_{ta}, A_{ta} A_{ta}^\top),$$

which indicates that $x(1), x(2) \sim N(\mu_t a_{ta}, a_{ta}^2)$. We also denote by

$$x' \triangleq x(1) + x(2) = [1, 1] \cdot x \sim N(2\mu_t a_{ta}, 4a_{ta}^2).$$

Then, we provide bound for the two situation.

**(a) The** $|a_{ta} - v_{ta}| \leq \delta_{1,t}$ **situation.** For any element in the cross term

$$e \in \mathbb{E}\left[ \frac{\partial^2 s_{\hat{\mu}, V_{ta}}}{\partial V_{ta}^2} (s_{\hat{\mu}, V_{ta}} - s_{\mu^*, A_{ta}}) \right],$$

we know that

$$|\mathbb{E}[e]| \leq 2|\mathbb{E}[\mu_t^2(1 - \tanh^2(\mu_t^\top V_{ta}^\top x))x(1)y(1)]|$$
$$+ 2|\mathbb{E}[\tanh''(\mu_t^\top V_{ta}^\top x)\mu_t^3(x(1)^2 + x(1)x(2))y(1)]| + 2|(\frac{1}{\sigma_t^2} - 1)\mathbb{E}[x(1)y(1)]|$$
$$\leq 2\mu_t^2\sqrt{\mathbb{E}[((1 - \tanh^2(\mu_t^\top V_{ta}^\top x))x(1))^2]}\sqrt{\mathbb{E}[y(1)^2]}$$
$$+ 2\mu_t^3\sqrt{\mathbb{E}[(\tanh''(\mu_t^\top V_{ta}^\top x)(x(1)^2 + x(1)x(2)))^2]}\sqrt{\mathbb{E}[y(1)^2]}$$
$$+ 2(\frac{1}{\sigma_t^2} - 1)\sqrt{\mathbb{E}[x(1)^2]}\sqrt{\mathbb{E}[y(1)^2]},$$

where the first inequality follows by the triangle inequality, and the second inequality follows by the Cauchy-Schwarz inequality. Then we give upper bounds on $\mathbb{E}[(\tanh'(\mu_t^\top V_{ta}^\top x))^2 x(1)^2]$, $\mathbb{E}[(\tanh''(\mu_t^\top V_{ta}^\top x)(x(1)^2 + x(1)x(2)))^2]$ and $\mathbb{E}[y^2]$ to obtain a total bound.

(i) Term $\mathbb{E}[(\tanh'(\mu_t^\top V_{ta}^\top x))^2 x(1)^2]$.

With the Cauchy-Schwarz inequality, we know that

$$\mathbb{E}[(\tanh'(\mu_t^\top V_{ta}^\top x))^2 x(1)^2] = \mathbb{E}[(1 - \tanh^2(\mu_t^\top V_{ta}^\top x))^2 x(1)^2]$$
$$\leq \sqrt{\mathbb{E}[\tanh'^4(\mu_t v_{ta}(x(1) + x(2)))]} \sqrt{\mathbb{E}[x(1)^4]}.$$

For the first component, we know that

$$\mathbb{E}_{x' \sim N(2\mu_t a_{ta}, 4a_{ta}^2)}[\tanh'^4(\mu_t v_{ta} x')]$$
$$\leq \int_0^\infty \tanh'^4(\mu_t v_{ta} x') \exp(-\frac{(x' - 2\mu_t a_{ta})^2}{8a_{ta}^2}) dx$$
$$\overset{x' = a_{ta} t}{=} a_{ta} \int_0^\infty \tanh'^4(\mu_t v_{ta} a_{ta} t) \exp(-\frac{(t - 2\mu_t)^2}{8}) dt$$
$$\leq a_{ta} \int_0^\infty \exp(-4\mu_t v_{ta} a_{ta} t) \exp(-\frac{(t - 2\mu_t)^2}{8}) dt$$
$$= a_{ta} \exp(4\mu_t v_{ta} a_{ta}(8\mu_t v_{ta}^2 - 2\mu_t)) \int_0^\infty \exp(-\frac{(t + 16\mu_t v_{ta}^2 - 2\mu_t)^2}{8}) dt$$
$$\leq a_{ta} \exp(4\mu_t^2 v_{ta} a_{ta}(8v_{ta}^2 - 2)).$$

Thus,

$$\mathbb{E}[(1 - \tanh^2(\mu_t^\top V_{ta}^\top x))^2 x(1)^2] \leq \sqrt{\mathbb{E}[\tanh'^4(\mu_t v_{ta}(x(1) + x(2)))]} \sqrt{\mathbb{E}[x(1)^4]}$$
$$\leq \sqrt{a_{ta}} \exp(4\mu_t^2 v_{ta} a_{ta}(4v_{ta}^2 - 1)) \sqrt{3 + 6\mu_t^2 + \mu_t^4 a_{ta}^2}$$
$$= \sqrt{3 + 6\mu_t^2 + \mu_t^4 a_{ta}^2} \sqrt{a_{ta}} \exp(4\mu_t^2 v_{ta} a_{ta}(4v_{ta}^2 - 1)),$$

where the second inequality follows the fact that $\mathbb{E}[x(1)^4] = (3 + 6\mu_t^2 + \mu_t^4) a_{ta}^4$.

We also know that $0 \leq (1 - \tanh^2(\mu_t^\top V_{ta}^\top x))^2 \leq 1$. As a result, we also can give another bound:

$$\mathbb{E}[(1 - \tanh^2(\mu_t^\top V_{ta}^\top x))^2 x(1)^2] \leq \mathbb{E}[x(1)^2] = (1 + \mu_t^2) a_{ta}^2$$

Hence, we can obtain that

$$\mathbb{E}[(1 - \tanh^2(\mu_t^\top V_{ta}^\top x))^2 x(1)^2]$$
$$\leq \min\{\sqrt{3 + 6\mu_t^2 + \mu_t^4 a_{ta}^2} \sqrt{a_{ta}} \exp(4\mu_t^2 v_{ta} a_{ta}(4v_{ta}^2 - 1)), (1 + \mu_t^2) a_{ta}^2\}.$$

(ii) Term $\mathbb{E}[(\tanh''(\mu_t^\top V_{ta}^\top x)(x(1)^2 + x(1)x(2)))^2]$.

For this term, we have that

$$\mathbb{E}[(\tanh''(\mu v_{ta}(x(1) + x(2)))(x(1)^2 + x(1)x(2)))^2] \leq \mathbb{E}[x(1)^2(x(1) + x(2))^2]$$
$$= \mathbb{E}[4x(1)^4] = 4(\mu_t^4 + 6\mu_t^2 + 3) a_{ta}^4,$$

where the first inequality holds because $0 \leq \tanh''^2(\mu v_{ta}(x(1) + x(2))) \leq 1$, the second and third equalities hold due to $x(1) = x(2)$ and $\mathbb{E}[x(1)^4] = (\mu_t^4 + 6\mu_t^2 + 3) a_{ta}^4$ respectively.

(iii) Term $\mathbb{E}[y^2]$. Recall that since we assume the latent parameter $\mu^*$ is perfectly learned by the pretraining phase $\hat{\mu} = \mu^*$, we do not especially distinguish $\hat{\mu}_t, \mu_t^*$ and $\mu_t$ in the proof of this section.

For $y$, we have that

$$y = s_{\hat{\mu}, V_{ta}} - s_{\mu^*, A_{ta}}$$

$$= \mu(V_{ta} \tanh(\mu_t^\top V_{ta}^\top x) - A_{ta} \tanh(\mu_t^\top A_{ta}^\top x)) - (V_{ta} V_{ta}^\top - A_{ta} A_{ta}^\top) x - \frac{1}{\sigma_t^2}(A_{ta} A_{ta}^\top - V_{ta} V_{ta}^\top) x$$

Let $A_{ta} = V_{ta} + \Delta$,

$$\mathbb{E}[y]$$

$$= \mathbb{E}[\mu_t(V_{ta} \tanh(\mu_t^\top V_{ta}^\top x) - A_{ta} \tanh(\mu_t^\top A_{ta}^\top x))] + \mathbb{E}[(1 - \frac{1}{\sigma_t^2})(V_{ta} \Delta^\top + \Delta V_{ta}^\top + \Delta \Delta^\top) x]$$

$$\leq \mathbb{E}[\mu_t(V_{ta} \tanh(\mu_t^\top V_{ta}^\top x) - A_{ta} \tanh(\mu_t^\top A_{ta}^\top x))] + (1 - \frac{1}{\sigma_t^2})(V_{ta} \Delta^\top + \Delta V_{ta}^\top + \Delta \Delta^\top) \mu_t A_{ta} .$$

We need to give the bound of $\mu_t(V_{ta} \tanh(\mu_t^\top V_{ta}^\top x) - A_{ta} \tanh(\mu_t^\top A_{ta}^\top x))$. Inspired by the Taylor's Theorem, we show $(x + \Delta x) \tanh(x + \Delta x) - x \tanh x$ can be bound by $K \Delta x$, where $K$ will be defined later.

$$f(x) = x \tanh(x)$$

$$f'(x) = \tanh(x) + x \cdot \text{sech}^2(x) = \tanh(x) + \frac{4x}{(\exp(x) + \exp(-x))^2} .$$

For the bound of $f'(x)$, we know that

$$|f'(x)| \leq |\tanh(x)| + \left| \frac{4x}{(\exp(x) + \exp(-x))^2} \right|$$

$$\leq \min\{1, |x|\} + \left| \frac{4x}{\exp(2x) + \exp(-2x) + 2} \right|$$

$$\leq \min\{1, |x|\} + \min\{|x|, \frac{2}{e}\} ,$$

where the first inequality holds because of the triangle inequality, the second inequality holds because $|\tanh(x)| \leq 1$ and $-x \leq \tanh(x) \leq x$. The third equality holds because

$$\left| \frac{4x}{\exp(2x) + \exp(-2x) + 2} \right| \leq x , \left| \frac{4x}{\exp(2x) + \exp(-2x) + 2} \right| \leq \frac{2}{e} .$$

For $y(1)$ in $y$, we have that

$$|y(1)| =$$

$$\left| v_{ta} \tanh(\mu_t(x(1) + x(2)) v_{ta}) - (v_{ta} + \delta) \tanh(\mu_t(x(1) + x(2))(v_{ta} + \delta)) + (1 - \frac{1}{\sigma_t^2})(2v_{ta}\delta + \delta^2) x(1) \right|$$

$$\leq \left| \frac{1}{\mu_t(x(1) + x(2))} \right| \left| (\min\{1, \mu_t(x(1) + x(2)) v_{ta}\} + \min\{\mu_t|x(1) + x(2)|v_{ta}, \frac{2}{e}\}) \delta \mu_t(x(1) + x(2)) \right|$$

$$+ \left| (1 - \frac{1}{\sigma_t^2})(2v_{ta}\delta + \delta^2) x(1) \right|$$

$$\overset{\delta < 1}{\leq} ((\mu(x(1) + x(2)) v_{ta} + \min\{\mu_t|x(1) + x(2)|v_{ta}, \frac{2}{e}\}) \delta + \left( \frac{1}{\sigma_t^2} - 1 \right)(2v_{ta} + 1) \delta x(1)$$

$$= \delta((\min\{1, \mu_t(x(1) + x(2)) v_{ta}\} + \min\{\mu_t|x(1) + x(2)|v_{ta}, \frac{2}{e}\} + \left( \frac{1}{\sigma_t^2} - 1 \right)(2v_{ta} + 1) x(1)) .$$

Recall that $x' = x(1) + x(2) \sim \mathcal{N}(2\mu_t a_{ta}, 4a_{ta}^2)$. For $\mathbb{E}[y(1)^2]$, we have that

$$\mathbb{E}[y(1)^2] \leq \mathbb{E}[(\delta((\min\{1, |\mu_t(x(1)+x(2))v_{ta}|\} + \min\{|\mu_t(x(1)+x(2))v_{ta}|, \frac{2}{e}\})]$$

$$+ \mathbb{E}[(1 - \frac{1}{\sigma_t^2})(2v_{ta}+1)x(1)))^2]$$

$$= (\mathbb{E}[(\min\{1, |\mu_t(x(1)+x(2))v_{ta}|\} + \min\{|\mu_t(x(1)+x(2))v_{ta}|, \frac{2}{e}\})^2]$$

$$+ (1 - \frac{1}{\sigma_t^2})^2(2v_{ta}+1)^2(1+\mu_t^2)v_{ta}^2\mathbb{E}[\min\{1, |\mu_t(x(1)+x(2))v_{ta}|\}$$

$$+ \min\{|\mu_t(x(1)+x(2))v_{ta}|, \frac{2}{e}\}](1 - \frac{1}{\sigma_t^2})(2v_{ta}+1)4v_{ta})\delta^2$$

$$\leq \left(4\mathbb{P}(|\mu_t v_{ta}x'| \geq \frac{2}{e}) + \mathbb{E}_{\mu_t v_{ta}x' < \frac{2}{e}}[4\mu_t^2 v_{ta}^2 x'^2] + (1 - \frac{1}{\sigma_t^2})^2(2v_{ta}+1)^2(1+\mu_t^2)v_{ta}^2\right.$$

$$+ (\mathbb{E}[2\mu_t v_{ta}x'] \quad + 2\mathbb{P}(|\mu_t v_{ta}x'| \geq \frac{2}{e}))(1 - \frac{1}{\sigma_t^2})(2v_{ta}+1)4v_{ta})\delta^2$$

$$\leq \left(4\mathbb{P}(|\mu_t v_{ta}x'| \geq \frac{2}{e}) + 16v_{ta}^2 a_{ta}^2 \mu_t^2(\mu_t^2+1) + (1 - \frac{1}{\sigma_t^2})^2(2v_{ta}+1)^2(1+\mu_t^2)v_{ta}^2\right.$$

$$+ (4\mu_t^2 v_{ta}a_{ta} + 2\mathbb{P}(|\mu_t v_{ta}x'| \geq \frac{2}{e}))(1 - \frac{1}{\sigma_t^2})(2v_{ta}+1)4v_{ta})\delta^2$$

$$\triangleq K^2\delta^2 \,,$$

where the first inequality follows by (i) dividing $\mu_t v_{ta}x'$ into two parts $\mu_t v_{ta}x' < 2/e$ and $\mu_t v_{ta}x' \geq 2/e$ (ii) $\min\{1, |\mu_t v_{ta}x'|\} = \min\{2/e, |\mu_t v_{ta}x'|\} = |\mu_t v_{ta}x'|$ when $\mu_t v_{ta}x' < \frac{2}{e}$ and the second inequality follows by $\mathbb{E}_{|\mu_t v_{ta}x'| < \frac{2}{e}}[\mu_t^2 v_{ta}^2 x'^2] \leq \mathbb{E}_x[\mu_t^2 v_{ta}^2 x'^2] = \mu_t^2 v_{ta}^2 a_{ta}^2(1+\mu_t^2)$.

For each element in the cross term $e \in \mathbb{E}\left[\frac{\partial^2 s_{\hat{\mu}, V_{ta}}}{\partial V_{ta}^2}(s_{\hat{\mu}, V_{ta}} - s_{\mu^*, A_{ta}})\right]$, it can be decompose into three term:

$$|\mathbb{E}[e]| \leq 2\left|\mathbb{E}[\mu_t^2(1 - \tanh^2(\mu_t^\top V_{ta}^\top x))x(1)y(1)]\right| + 2\left|(\frac{1}{\sigma_t^2} - 1)\mathbb{E}[x(1)y(1)]\right|$$

$$+ 2\left|\mathbb{E}[\tanh''(\mu_t^\top V_{ta}^\top x)\mu_t^3(x(1)^2 + x(1)x(2))y(1)]\right| \,.$$

For the first term, we have that

$$2\left|\mathbb{E}[\mu_t^2(1 - \tanh^2(\mu_t^\top V_{ta}^\top x))x(1)y(1)]\right|$$

$$\leq K\delta\left(2\mu_t^2\sqrt{\sqrt{\mu_t^4 + 6\mu_t^2 + 3}a_{ta}^2\sqrt{a_{ta}}\exp(4\mu_t^2 v_{ta}a_{ta}(4v_{ta}^2-1))}\right) \,.$$

For the second term, we have that

$$2\left|\mathbb{E}[\tanh''(\mu_t^\top V_{ta}^\top x)\mu_t^3(x(1)^2 + x(1)x(2))y(1)]\right| \leq K\delta\left(2\mu_t^3\sqrt{4(\mu_t^4 + 6\mu_t^2 + 3)a_{ta}^4}\right) \,.$$

For the third term, we have that

$$2\left|(\frac{1}{\sigma_t^2} - 1)\mathbb{E}[x(1)y(1)]\right| \leq K\delta\left(2(\frac{1}{\sigma_t^2} - 1)\sqrt{1+\mu_t^2}a_{ta}\right) \,.$$

Combined with the bound for these three term, we have that

$$|\mathbb{E}[e]|$$

$$\leq 2a_{ta}\mu_t^2 K\delta\times$$

$$\left(a_{ta}^{\frac{1}{4}}\sqrt[4]{\mu_t^4 + 6\mu_t^2 + 3}\exp(2\mu_t^2 v_{ta}a_{ta}(4v_{ta}^2-1)) + 2\sqrt{\mu_t^4 + 6\mu_t^2 + 3}\mu_t a_{ta} + (\frac{1}{\sigma_t^2} - 1)\sqrt{1+\mu_t^2}\right)$$

$$= KC_4\delta \,, \tag{15}$$

where $\delta \in |\Delta| = |V_{ta} - A_{ta}| \geq 0$.

Now we focus on the Hessian matrix. Let $H$ be the $2 \times 2$ Hessian matrix, $\gamma \triangleq KC_4\delta$,

$$\alpha \triangleq \mathbb{E}_{x \sim p_{data}} \left[ p^2 + pqx^\top V_{ta} + \frac{q^2 \|x\|^2 \|V_{ta}\|^2}{2} - \frac{q}{2} \sqrt{\|x\|^2 \|V_{ta}\|^2 \left(4p^2 + 4pqx^\top V_{ta} + q^2 \|x\|^2 \|V_{ta}\|^2\right)} \right],$$

where $p = \tanh(\hat{\mu}_t^\top V_{ta}^\top x_t)\hat{\mu}_t + \left(\frac{1}{\sigma_t^2} - 1\right) V_{ta}^\top x_t$ and $q = (1 - \tanh^2(\hat{\mu}_t^\top V_{ta}^\top x_t))\hat{\mu}_t^\top \hat{\mu}_t + \frac{1}{\sigma_t^2} - 1 > 0$.

As we defined before, we can divide $H$ into two parts $H_1$ and $H_2$:

$$H_1 = \begin{bmatrix} h_1 & 0 \\ 0 & h_1 \end{bmatrix}, h_1 \geq \alpha - \gamma$$

$$H_2 = \begin{bmatrix} h_2 & h_2 \\ h_2 & h_2 \end{bmatrix}, h_2 \geq \alpha' - \gamma,$$

where $\alpha$ and $\alpha'$ is determined in Lemma 6.1. Thus, if $h_1 > 0$ and $h_2 \geq 0$, the Hessian matrix $\frac{\partial^2 \mathcal{L}_{\text{SM},t}^{\text{few}}}{\partial V_{ta}^2}$ is $2(\alpha - \gamma)$-positive definite.

In our example (Example 11), $\mu_t = 4 \exp(-2)$, $a_{ta} = 0.12$, $\sigma_t = \sqrt{1 - \exp(-4)}$. $\mathbb{P}(|\mu_t v_{ta} x'| \geq \frac{2}{e}) \leq 1 \times 10^{-20} \approx 0$.

Then, we know that when $\delta \leq 0.02$ ($v_{ta} \in [0.1, 0.14]$) $\alpha - \gamma \geq 0$, and

$$h_2 \geq \mathbb{E}_{x(1) \sim \mathcal{N}(\mu_t a_{ta}, a_{ta}^2)}[2(1 - \tanh^2(0.28\mu_t x(1)))^2 \mu_t^2 v_{ta}^2 x(1)^2] - \gamma \geq 0.$$

**The $v_{ta} \geq a_{ta} + \delta_{1,t}$ situation.** When $v_{ta} \geq a_{ta}$, we will prove that the cross term is semi-positive definite in a large region. If $v_{ta} > a_{ta}$ and $\mu_t = \mu^\star$, we can get $x^\top y \geq 0$ and $(1 - \tanh^2(\mu^\top V_{ta}^\top x))x^\top y \geq 0$:

$$x^\top y = x^\top (s_{\hat{\mu}, V_{ta}} - s_{\mu^*, A_{ta}})$$

$$= x^\top V_{ta} \tanh(\mu_t V_{ta}^\top x)\mu_t - x^\top A_{ta} \tanh(\mu_t^\star A_{ta}^\top x)\mu_t^\star + \left(\frac{1}{\sigma_t^2} - 1\right) x^\top (V_{ta} V_{ta}^\top - A_{ta} A_{ta}^\top)x$$

$$\geq x^\top A_{ta} \tanh(\mu_t A_{ta}^\top x)\mu_t - x^\top A_{ta} \tanh(\mu_t^\star A_{ta}^\top x)\mu_t^\star + \left(\frac{1}{\sigma_t^2} - 1\right) x^\top (A_{ta} A_{ta}^\top - A_{ta} A_{ta}^\top)x$$

$$= 0,$$

where the inequality holds because $x^\top V_{ta} \tanh(\mu_t V_{ta}^\top x)\mu$ is even, monotonically increasing if $V_{ta}^\top x \geq 0$ and $V_{ta}^\top x \geq A_{ta}x$.

Then, we have that

$$tr(xy^\top) = tr(y^\top x) = tr(x^\top y) \geq 0$$

and

$$\text{Rank}(xy^\top) \leq \text{Rank}(x) = 1.$$

We also know that $1 - \tanh^2(\mu_t V_{ta}^\top x) \geq 0$, which indicates $(1 - \tanh^2(\mu_t V_{ta}^\top x))xy^\top$ is semi-positive definite.

Recall that the cross term has the following form

$$M = \mathbb{E}\left[ \frac{\partial^2 s_{\hat{\mu}, V_{ta}}}{\partial V_{ta}^2} \left(s_{\hat{\mu}, V_{ta}} - s_{\mu^*, A_{ta}}\right)\right]$$

$$= \mathbb{E}[(1 - \tanh^2(\mu_t^\top V_{ta}^\top x))\mu_t^\top \mu_t x^\top y I_2 + (1 - \tanh^2(\mu_t^\top V_{ta}^\top x))\mu_t^\top \mu_t xy^\top$$

$$- 2\tanh(\mu^\top V_{ta}^\top x)(1 - \tanh^2(\mu_t^\top V_{ta}^\top x))\mu_t^\top \mu_t \mu_t x^\top yx V_{ta}^\top + 2\left(\frac{1}{\sigma_t^2} - 1\right) x^\top y I_2].$$

Then, we define the following two matrix: $M = M' + M_1$, where

$$M' = (1 - \tanh^2(\mu_t^\top V_{ta}^\top x))\mu_t^\top \mu_t xy^\top + 2\left(\frac{1}{\sigma_t^2} - 1\right) x^\top y I_2, \tag{16}$$

and

$$M_1 = (1 - \tanh^2(\mu_t^\top V_{ta}^\top x))\mu_t^2 x^\top y I_2 - 2\tanh(\mu_t^\top V_{ta}^\top x)\mu_t x V_{ta}^\top(1 - \tanh^2(\mu_t^\top V_{ta}^\top x))\mu_t^2 x^\top y\,.$$

We know that

$$M_1[1,1] + M_1[1,2] = (1 - \tanh^2(\mu_t V_{ta}^\top x))\mu_t^2 x^\top y(1 - 4\tanh(\mu_t V_{ta}^\top x)\mu_t v_{ta}x(1))$$

$$\mathbb{E}[M_1[1,1] + M_1[1,2]]$$
$$\geq \mathbb{E}[(1 - \tanh^2(\mu_t V_{ta}^\top x))x^\top y(1 - 4\tanh(\mu_t V_{ta}^\top x)\mu_t ax(1)]$$
$$= \mathbb{E}[(1 - \tanh^2(\mu_t V_{ta}^\top x)x^\top y] - \mathbb{E}[4(1 - \tanh^2(\mu_t V_{ta}^\top x))x^\top y\tanh(\mu_t V_{ta}^\top x(1))\mu_t v_{ta}x(1))]\,.$$

Then, we discuss each component in the following part. For the first term, we know that

$$\mathbb{E}[(1 - \tanh^2(\mu_t V_{ta}^\top x))x^\top y] \geq \mathbb{E}[(1 - \tanh^2(\mu_t x(1)))x^\top y] \stackrel{\triangle}{=} u_1(v_{ta}, t)\,. \tag{17}$$

For the second term, we know that

$$-\mathbb{E}[4(1 - \tanh^2(\mu_t V_{ta}^\top x))x^\top y\tanh(\mu_t V_{ta}^\top x(1))\mu_t v_{ta}x(1))]$$
$$\geq -\mathbb{E}[4x^\top y\tanh(\mu_t V_{ta}^\top x(1))\mu_t v_{ta}x(1))] \stackrel{\triangle}{=} u_2(v_{ta}, t)\,. \tag{18}$$

We know that $u_1(v_{ta}, t)$ increases with $v_{ta}$ increasing while $u_2(v_{ta}, t)$ decreases with $v_{ta}$ increasing. We also know that when $v_{ta,t} = a_{ta,t} + \delta_{1,t}$, $u_1(v_{ta}, t) + u_2(v_{ta}, t) > 0$, which indicates there exists an area $v_{ta,t} \in [a_{ta,t} + \delta_{1,t}, a_{ta,t} + \delta_{2,t}]$ that $M_1[1,1] + M_1[1,2] \geq 0$.

Thus,

$$\mathbb{E}[M_1[1,1]] = \mathbb{E}[M_1[2,2]] > 0, \mathbb{E}[M_1[1,2]] = \mathbb{E}[M_1[2,1]] < 0\,,$$

and

$$|\mathbb{E}[M_1]| = (\mathbb{E}[M_1[1,1]])^2 - (\mathbb{E}[M_1[1,2]])^2$$
$$= (\mathbb{E}[M_1[1,1]] + \mathbb{E}[M_1[1,2]])(\mathbb{E}[M_1[1,1]] - \mathbb{E}[M_1[1,2]]) > 0\,.$$

Then we know that

$$\mathbb{E}_x[(1 - \tanh^2(\mu_t^\top V_{ta}^\top x))\mu_t^2 x^\top y I_2 - 2\tanh(\mu_t^\top V_{ta}^\top x)\mu_t x V_{ta}^\top(1 - \tanh^2(\mu_t^\top V_{ta}^\top x))\mu_t^2 x^\top y]$$

is semi-positive definite. Then, the proof is finished.

To make a clearer discussion, we use the setting of Example 11 to show the interval of $[a_{ta} + \delta_{1,t}, a_{ta} + \delta_{2,t}]$.

$v_{ta} \in [0.14, 0.25]$

$$u_1(0.14) \approx 0.00023 > 0.0002$$
$$u_2(0.28) \leq 4 \times 10^{-5} < f(0.14)$$

$v_{ta} \in [0.25, 0.4]$

$$u_1(0.25) \approx 0.0021 > 0.002$$
$$u_2(0.4) \leq 1.4 \times 10^{-4} < f(0.25)$$

$v_{ta} \in [0.4, 0.5]$

$$u_1(0.4) \approx 0.0064 > 0.006$$
$$u_2(0.5) \leq 0.00034 < f(0.4)$$

Hence, we can have $\mathbb{E}[M_1(1,1) + M_1(1,2)] > 0$ when $v_{ta} \in [0.14, 0.5]$. ∎

Before proving our convergence guarantee, we first previous convergence lemma.

**Lemma F.4** (Convergence Lemma). *Let $\phi$ be locally $\mu$-strongly convex and $L_m$-smooth, if $\eta_t = \eta = \frac{2}{\mu + L_m}$, $\kappa = \frac{L_m}{\mu}$, and $x^* \in \arg\min_{x \in \mathcal{X}} \phi(x)$, then*

$$\left\| x^t - x^* \right\|_2 \leq \left( \frac{\kappa - 1}{\kappa + 1} \right)^t \left\| x^{(0)} - x^* \right\|_2 .$$

After that, we provide our convergence guarantee for few-shot diffusion models with a great pretraining.

**Theorem 6.4.** *Assume Assumption 3.1, 5.1, $\hat{\mu} = \mu^*$ and $\delta_{1,t}, \delta_{2,t}$ satisfy **Condition 1**. Considering score matching function $\mathcal{L}_{\mathrm{SM},t}^{\mathrm{few}}$. When $v_{ta}^{(0)} \in \{[a_{ta} - \delta_{1,t}, a_{ta} + \delta_{2,t}] \cup [-a_{ta} - \delta_{2,t}, -a_{ta} + \delta_{1,t}]\}$, using gradient descent with learning rate $\eta = 1/(2\alpha + \zeta)$, with $\kappa = (\alpha + \gamma + \zeta)/(\alpha - \gamma)$, we have*

$$\left\| V_{ta}^{(k)} V_{ta}^{(k)\top} - A_{ta} A_{ta}^\top \right\|_F \leq \left( \frac{\kappa - 1}{\kappa + 1} \right)^k (2a_{ta} + \delta_{2,t}) |v_{ta}^{(0)} - a_{ta}| .$$

**Proof.** First we prove that there exists $L_m > 0$, such that the objective function is $L_m$-smooth. In this work, we take the maximum eigenvalue of the hessian matrix to be $L_m$.

$$\mathbb{E}\left[ \frac{\partial^2 \mathcal{L}_{\mathrm{SM}}^{\mathrm{few}}}{\partial V_{ta}^2} \right] = 2\mathbb{E}\left[ (\frac{\partial^2 s_{\hat{\mu}, V_{ta}}}{\partial V_{ta}^2})^\top (s_{\hat{\mu}, V_{ta}} - s_{\mu^*, A_{ta}}) \right] + 2\mathbb{E}\left[ \left( \frac{\partial s_{\hat{\mu}, V_{ta}}}{\partial V_{ta}} \right)^\top \left( \frac{\partial s_{\hat{\mu}, V_{ta}}}{\partial V_{ta}} \right) \right]$$

Based on our analysis of the hessian matrix, we can divide the matrix into two parts: $\begin{bmatrix} \lambda_1 & 0 \\ 0 & \lambda_1 \end{bmatrix}$ and $\begin{bmatrix} \lambda_2 & \lambda_2 \\ \lambda_2 & \lambda_2 \end{bmatrix}$.

We first analyze the property of $\begin{bmatrix} \lambda_1 + \lambda_2 & \lambda_2 \\ \lambda_2 & \lambda_1 + \lambda_2 \end{bmatrix}$, and then give the bound of $\lambda_1$ and $\lambda_2$.

$$\left| \lambda I_2 - \begin{bmatrix} \lambda_1 + \lambda_2 & \lambda_2 \\ \lambda_2 & \lambda_1 + \lambda_2 \end{bmatrix} \right| = 0 \Rightarrow (\lambda - \lambda_1)(\lambda - \lambda_1 - \lambda_2) = 0 ,$$

which indicates $\lambda = \lambda_1$ or $\lambda = \lambda_1 + \lambda_2$. Thus, if $\lambda_1 > 0$, we can choose $L_m = \lambda_1 + |\lambda_2|$

According to our analysis on before, $\forall e \in \mathbb{E}[(\frac{\partial^2 s_{\hat{\mu}, V_{ta}}}{\partial V_{ta}^2})^\top (s_{\hat{\mu}, V_{ta}} - s_{\mu^*, A_{ta}})], |e| \leq v$

Next we analyze $\mathbb{E}\left[ \left( \frac{\partial s_{\hat{\mu}, V_{ta}}}{\partial V_{ta}} \right)^\top \left( \frac{\partial s_{\hat{\mu}, V_{ta}}}{\partial V_{ta}} \right) \right]$ and have the following form:

$$\mathbb{E}\left[ \left( \frac{\partial s_{\hat{\mu}, V_{ta}}}{\partial V_{ta}} \right)^\top \left( \frac{\partial s_{\hat{\mu}, V_{ta}}}{\partial V_{ta}} \right) \right]$$

$$= \mathbb{E}[\tanh^2(\mu_t^\top V_{ta}^\top x)\mu_t^\top \mu_t I_2] + \mathbb{E}[(1 - \tanh^2(\mu_t^\top V_{ta}^\top x))^2 \mu_t^\top \mu_t V_{ta} x^\top x V_{ta}^\top]$$

$$+ 2(\frac{1}{\sigma_t^2} - 1)^2 ((1 + \mu_t^2) A_{ta} A_{ta}^\top V_{ta} V_{ta}^\top + (1 + \mu_t^2) V_{ta}^\top A_{ta} A_{ta}^\top V_{ta})$$

$$+ \mathbb{E}[2(1 - \tanh^2(\mu_t^\top V_{ta}^\top x)) \tanh(\mu_t^\top V_{ta}^\top x)\mu_t \mu_t^\top \mu_t V_{ta} x^\top]$$

$$+ 2\mathbb{E}[(\frac{1}{\sigma_t^2} - 1) \tanh(\mu_t^\top V_{ta}^\top x)\mu_t(x V_{ta}^\top + V_{ta}^\top x I_2)]$$

$$+ \mathbb{E}(1 - \tanh^2(\mu_t^\top V_{ta}^\top x)) \left( \frac{1}{\sigma_t^2} - 1 \right) \mu_t^\top \mu_t (V_{ta} x^\top x V_{ta}^\top + V_{ta} x^\top V_{ta}^\top x)$$

$$+ \mathbb{E}(1 - \tanh^2(\mu_t^\top V_{ta}^\top x)) \left( \frac{1}{\sigma_t^2} - 1 \right) \mu_t^\top \mu_t (x^\top V_{ta} x V_{ta}^\top + x^\top V_{ta} V_{ta}^\top x)$$

$$= \begin{bmatrix} \alpha & 0 \\ 0 & \alpha \end{bmatrix} + \begin{bmatrix} \zeta & \zeta \\ \zeta & \zeta \end{bmatrix} ,$$

where

$$\zeta = \mathbb{E}_{x(1)\sim\mathcal{N}(\mu_t a_{ta}, a_{ta}^2)}\left[2(1 - \tanh^2(2\mu_t v_{ta} x(1)))^2 \mu_t^2 v_{ta}^2 x(1)^2\right.$$

$$+ 2(1 - \tanh^2(2\mu_t v_{ta} x(1)))\tanh(2\mu_t v_{ta} x(1))\mu_t^3 v_{ta} x(1)$$

$$+ 2\left(\frac{1}{\sigma_t^2} - 1\right)\tanh(2\mu_t v_{ta} x(1))\mu_t a x(1)$$

$$\left. + \left(\frac{1}{\sigma_t^2} - 1\right)(1 - \tanh^2(2\mu_t v_{ta} x(1)))6\mu_t^2 v_{ta}^2 x(1)^2 + \left(\frac{1}{\sigma_t^2} - 1\right)^2 4v_{ta}^4 x(1)^4\right]. \quad (19)$$

For the $\zeta$, we have the following bound:

$$\zeta \le \mathbb{E}\left[2\mu_t^2 v_{ta}^2 x(1)^2 + \left(\frac{1}{\sigma_t^2} - 1\right)^2 4v_{ta}^4 x(1)^4 + 4\mu_t^4 v_{ta}^2 x(1)^2\right]$$

$$+ 4\mathbb{E}\left[\left(\frac{1}{\sigma_t^2} - 1\right)\mu_t^4 v_{ta}^2 x(1)^2 + 6\left(\frac{1}{\sigma_t^2} - 1\right)\mu_t^2 v_{ta}^2 x(1)^2\right]$$

$$= \mu_t^2 v_{ta}^2 a_{ta}^2 (1 + \mu_t^2)[2 + 4\mu_t^2 + 4\left(\frac{1}{\sigma_t^2} - 1\right)\mu_t^2 + 6\left(\frac{1}{\sigma_t^2} - 1\right)]$$

$$+ 4\left(\frac{1}{\sigma_t^2} - 1\right)^2 (\mu_t^4 + 6\mu_t^2 + 3)v_{ta}^4 a_{ta}^4.$$

Thus, we can take $L_m = 2(\alpha + \gamma + \zeta)$. Let $\kappa = \frac{L_m}{\alpha - \gamma}$, $\eta = \frac{2}{2(\alpha-\gamma)+2(\alpha+\gamma+\zeta)} = \frac{1}{2\alpha+\zeta}$ and $A_{ta} \in \operatorname{argmin}_{V_{ta}\in\mathcal{Q}(\hat\mu)} L_{\text{SM},t}^{\text{few}}$, then

$$\left\|V_{ta}^{(k)}V_{ta}^{(k)\top} - A_{ta}A_{ta}^\top\right\|_F \le \left(\frac{\kappa-1}{\kappa+1}\right)^k (2a_{ta} + \delta_{2,t})|v_{ta}^{(0)} - a_{ta}|.$$

∎