# OpenReview forum: "Evaluating the Role of Great Pre-trained Diffusion Models in Few-shot Phase: Warm-up and Acceleration"
_ICLR.cc/2026/Conference — Submitted to ICLR 2026_

### Official Review · Reviewer_458S · 2025-10-26

**Soundness:** 3
**Presentation:** 3
**Contribution:** 1
**Rating:** 2
**Confidence:** 2

**Summary:**

This paper addresses an important and empirically observed phenomenon: the rapid convergence of few-shot diffusion models like DreamBooth. The authors claim to be the first to provide a theoretical optimization guarantee for this *fast convergence*. The core argument is that a "great" pre-trained model provides a "warm-up" by transforming the few-shot optimization landscape into one that is *strongly convex*.

**Strengths:**

Idea is intriguing as it aligns well with the empirical success of methods like LoRA, which also involve fine-tuning linear adaptations.

**Weaknesses:**

1. The paper's theoretical guarantee hinges on an extremely simplified experimental setting: the data is assumed to lie on a linear subspace, and the shared latent distribution is modeled as a simple 2-mode Gaussian Mixture Model (GMM). While this "toy problem" setting certainly makes the theoretical analysis tractable (i.e., allows for a closed-form score function), it is a far from the complex, high-dimensional, and non-linear manifolds of real-world image data. It is highly questionable whether the finding of strong convexity in this trivial setting holds any meaningful implication for the optimization of large-scale models like Stable Diffusion on real images.

2. The proposed problem can be solved from existing literature: The primary concern is whether the paper's score-matching-based analysis is necessary at all. The objective function for diffusion models, Denoising Score Matching ($\mathcal{L}_{DSM}$), is essentially equivalent to a weighted **Mean Squared Error (MSE)** problem, where the model is trained to predict either the original data or the added noise from a corrupted input.* Therefore, this entire problem can be **re-formulated**: it is not about the convergence of a "Score Matching" function, but about the fine-tuning of a **pre-trained regression model on an MSE loss**.* There is already a vast body of existing literature in transfer learning theory that explains why and how the fine-tuning of pre-trained regression models converges quickly (e.g., good feature extraction, flat minima, etc.).The authors fail to justify **why this existing body of literature is insufficient** to explain their observations. They must clearly articulate what *fundamentally new insight* their complex, score-function-based analysis provides that could not have been derived by applying standard transfer learning theory to the equivalent MSE problem.

**Questions:**

See weakness.

---

> ### Author Response · Authors · 2025-12-03
>
> **Weakness 1: Simplified Assumption**
>
> We thank the reviewers for their valuable comments, and based on the reviewer's insightful feedback, we have successfully extended our proofs to the **general K-mode setting**.We provide a detailed K-mode analysis in the global response now.
>
> **Weakness 2:The Necessity of Score-Matching Analysis**
>
> We thank the reviewer for this insightful observation. While the objective function is indeed MSE, the few-shot fine-tuning parameters are highly depends on the structure of diffusion score, which indicates that applying generic regression theory would miss the structural decomposition and the specific failure modes in diffusion models. We respectfully disagree that generic transfer learning theory for regression is sufficient to explain the unique optimization dynamics of few-shot diffusion models. Our analysis starts from the empirical observations and explain the empirical observations (the influence of good and bad pretrained models) from the theoretical perspective.
>
> **(a) NN structure from the diffusion models.**
> In this work, we adopt the linear subspace structure (which matches the property of real world data) and naturally determined the encoder-latent score-decoder structure score function. Existing literature does not address how this specific "linear subspace + fixed latent score" decomposition behaves during optimization. By focusing on this structure, we prove that the few-shot phase is not just generic fine-tuning, but specifically the alignment of linear parameters to a complex, multi-modal latent distribution (modeled as a GMM). This structural insight is absent in general MSE theory.
>
> **(b) The optimization analysis for bad and good pretrianed.**
>
> *The warm-up and convergence guarantee with good pretrained model.*
>
> Our work goes a step further by **quantifying** this process. We provide strict bounds on the convergence rate.
>
> Specifically, **Theorem 6.4** provides the first explicit convergence guarantee for few-shot diffusion models, deriving the convergence rate $\kappa$ as a function of the initialization area and the latent structure. This explains the "acceleration" not just as "good features," but as a fundamental geometric simplification of the loss landscape caused by the warm-up of the latent score.
>
> *The influence of bad pretrained model.*
>
> Finally, generic MSE theory interprets failure simply as "high error." It fails to explain specific *generative* failure modes. Our score-based analysis explains **why** bad pre-training leads to specific phenomena like the "memory phenomenon" (where the model generates training data instead of new concepts).
>
> In **Theorem 5.5**, we analyze the gradient of the score function (involving non-linear $\tanh$ terms) and prove that a misaligned latent prior leads to vanishing gradients. This mechanical explanation links the optimization difficulty directly to the semantic collapse of the generative model, a level of granularity that standard regression theory cannot achieve.

---

### Official Review · Reviewer_PsXu · 2025-10-31

**Soundness:** 2
**Presentation:** 2
**Contribution:** 2
**Rating:** 2
**Confidence:** 4

**Summary:**

The paper focuses on few-shot adaptation of pre-trained diffusion models, studying the theoretical aspect of fast adaptation of diffusion models with only a few optimization steps. The observations they found include: if the pre-trained model is not good, few-shot adaptation cannot find the groundtruth parameters, yielding a bad model but with small gradients. They contrasted their work with several previous related works aiming for few-shot diffusion model guarantees, where they positioned their paper to explain about a few optimization steps, which was not done before. They proved: a few-shot adaptation with a great pre-trained model converges to the groundtruth parameters with convergence guarantee, with few optimization steps.

**Strengths:**

- The problem of theoretical analysis of the few-shot diffusion fine-tuning they tried to tackle looks interesting.

**Weaknesses:**

- First, the use of the term "great" in the title sounds too informal. Use a more objective and formal term instead.

- Sec.4 about the failure of few-shot fine-tuning of bad pre-trained (overfit, underfit) models looks too obvious. Do you need to do this?

- Sec. 5 aims to show with bad pre-trained models the fine-tuning cannot find the groundtruth parameters by extending (Yang et al. 2024) from gaussian $q_z$ to a mixture of gaussians.

- There are a bunch of assumptions made that sound overly simplified and unrealistic, such as the mixture of Gaussians with the means exponentially decaying with time and opposite signs; the mixture of two isotropic Gaussians (not shown to be extended to general mixtures), and the particular choice of the one-layer tanh score neural network assumption.

- In Thm 5.5, they show the gradient is bounded, but it all depends on some unknown constants that are unknown. So it is a logical leap to argue that the gradient is small.

- The assumption 3.1 says that the latent is shared between pre-trained data and the target data distributions, and they are linearly related to the latent. Let's say this assumption is true considering that the same assumption was made in (Yang et al. 2024). But the follow-up assumption 5.1 of the mixture of two gaussians with the same isotropic covariances and exponentially decaying means with opposite signs, looks too contrived and simplified. Also they rely on the particular (simple one-hidden tanh layer networks) score neural network defined in the unlabelled equation right below after Eq.(3). All subsequent theorems are based on these assumptions, and considering that the paper is aiming to become a theoretical paper with lack of strong empirical evidence, it is not a solid paper.

- Sec. 6, the argument about the strong convex few-shot optimization, is also relying on those particular (unrealistic and overly simplified) assumptions.

- In theorem 6.4, the convergence also depends on the scale of the $\kappa$ value. Eg, if $\kappa >> 1$, then it is hard to conclude that you can attain optimal values in just a few steps. In other words, the practical significance of this theoretical result is doubtful.They did some simulations in Table 2, but it looks overly contrived.

- Overall, reading this paper gives me an impression that they are building a sandcastle in the air on top of unrealistic and overly simplified assumptions. Any empirical evidence, eg, how SD1.4 fits in their assumptions, would make their argument convincing.

**Questions:**

See questions in the weakness section.

---

> ### Author Response · Authors · 2025-12-03
>
> **Weakness 1:How to get "Bad" pre-trained models**
> We thank the reviewers for their valuable comments. We agree that in practical applications, "bad" models are usually due to poor data quality or insufficient training, rather than intentional overfitting.
>
> However, we want to clarify that the "dog image overfitting" experiment was intended as an **extreme** simulation to visualize the theoretical concept of the "memory phenomenon" discussed in Section 4. We experimentally validated our theoretical findings that biased initialization may create a strong local minimum that is difficult to escape during fine-tuning with few samples, leading to failure in learning new concepts (e.g., cats). This can serve as an explanation for failed fine-tuning cases in reality.
>
> **Weakness 4: General K-mode latent**
>
> We thank the reviewers for their valuable comments, and based on the reviewer's insightful feedback, we have successfully extended our proofs to the **general K-mode setting**.We provide a detailed K-mode analysis in the global response now.
>
> **Weakness 5: Small Gradient**
>
> In our theoretical analysis, we prove that the gradient can be bounded by **Const \*$\epsilon$**. Since the Const is O(1), the gradient magnitude is small when $\epsilon$ is small (**O($\epsilon$)**), which makes it difficult for the optimizer to escape the local minimum.
>
> **Weakness 6 && Weakness7: Network Architecture**
>
> We thank the reviewer for the critical assessment of our theoretical assumptions. We understand the concern that the specific network structure might appear simplified.
>
> We respectfully clarify that this architecture is **not** arbitrarily designed for our specific proofs. Instead, it is strictly adopted from the established theoretical framework for learning Gaussian Mixtures by Shah et al. [1] For a mixture of Gaussians, the score function has a specific **closed-form analytical structure**. The network used in our paper is the **optimal parameterization** required to approximate this score structure. Therefore, it is the most rigorous choice for analyzing optimization dynamics in multi-modal settings, rather than a simplified heuristic.
>
>
>
> **Weakness 8: Condition Number $\kappa$**
>
> We thank the reviewer for this insightful comment regarding the condition number and the validity of our simulations. We respectfully clarify that we attempt to provide deeper insights from a theoretical perspective based on experimental observations, and our simulations are designed to verify the fundamental shift in the optimization landscape.
>
> Considering the low dimension of the latent space, theoretically, the MoG distribution can reflect the latent space state quite well after being extended to k modes.The practical value of this result is to explain the order-of-magnitude speedup observed in applications (from 500k steps in pre-training to <1k in few-shot).
>
> [1]Kulin Shah, Sitan Chen, Adam Klivans. Learning Mixtures of Gaussians Using the DDPM Objective.Advances in Neural Information Processing Systems 37 (2023).

---

### Official Review · Reviewer_keJV · 2025-11-01

**Soundness:** 3
**Presentation:** 3
**Contribution:** 3
**Rating:** 6
**Confidence:** 2

**Summary:**

The authors provide theoretical backing and insights to the question why few-shot diffusion models require only a couple of optimisation steps, and how the convergence behaviour as well as resulting performance after adaptation is affected by different ‘levels of quality’ of the pre-training (i.e. initialisation) stage. The presented theoretical insights are additionally motivated and supported through a few experiments to reiterate the well-known empirical findings.

**Strengths:**

**Originality & Significance:**
- The authors do a good job in outlining why their analyses matter, and that this work tries to provide theoretical backing for empirically observed insights
- The theoretical analyses and backing are first motivated through some empirical analyses with current popular models, which provides a nice basis to ‘grasp’ what the authors aim to theoretically show later

**Quality:**
- Their work is placed well within related efforts, and the specific gap the authors tackle is clearly presented and justified
- Combination of the empirical and theoretical angle regarding what a ‘good/great’ and ‘bad’ pretrained model is opens the insights up to a wider audience, especially given the provides visuals

**Clarity:**
- The work is mostly well written and easy to follow, and a good mix of illustrations, text and equations

**Weaknesses:**

**Major:**
- Interesting fact/observation regarding the difference of the sampling process that is used within the diffusion model unfortunately remains unexplored, and no comment is made how/whether this could affect the statements/conclusions drawn regarding adaptation capabilities; see questions.
- Experimental results provide limited new insights: While I understand the intention to (again) shot that badly trained models adapt worse than well-trained ones, the experimental side of the results seems very expected and yields little-to-no surprise: The fact that very overfitted and/or undertrained models don’t adapt well/fast is quite well known across the few-shot learning community, and several works have been tailored to combat this (e.g. avoiding supervision collapse in works like Doersch et al., NeurIPS2020 or Hiller et al., NeurIPS2022, etc.)
- Some remaining questions on how the presented simplified setup relates (and generalises) to the usually more complex, real-world experiments – see questions.

**Minor:**
- Consistency in formatting of formulas would be improved in places, e.g. eq in l.191 keeps the $\sigma^2$ in the denominator of each term, whereas the closely-related eq in l.200 uses a pre-factor of $1/sigma^2$; Keeping them in the same form would make it easier for the reader so directly see the (almost) identical formulations
- Choice of words could be improved to avoid confusion in some settings, see questions.
- (minor suggestion) Structure of manuscript could be improved, as many sections have only one subsection (e.g. 3.1, 5.1 but no second); I’d recommend to either make it 3.1 and 3.2, or simply keep it all in the main one (might be down to preference though)

**Questions:**

**Main Questions:**
- The authors present their ‘interesting observation’ regarding the difference in adaptation quality that results from a deterministic sampling process (in SD3) vs. the more stochastic process in SD1.4. This observation is unfortunately only mentioned in the text with results shown in Table 1, but stays entirely unexplored beyond this: I’d be quite curious whether it’s possible to include this into their analysis and/or provide any theoretical backing/background for this?
$\rightarrow$ I assume this is due to the additional energy that is introduced to the diffusion process when adding additional noise during the sampling process, which from an empirical standpoint provides more chances for the model to correct initial errors; and in some sense to then ‘cover more ground’ during generation time
$\rightarrow$ However, given that the authors attempt to provide insights from a theoretical standpoint, I’d like to hear some comments on their thoughts, and how / whether this would influence their current derivations/analyses & conclusions!
$\rightarrow$ Is there a way to quantify/formulate this ability of apparently being able to compensate for worse pretraining?
- The authors show the difference in the loss landscape (re. convexity) via the sign of the Eigenvalues, and go on to deduct the convergence properties;
$\rightarrow$ However, although a problem can be convex and therefore converge eventually, the actual ‘convergence speed’ isn’t necessarily ‘fast’ or ‘in few steps’ when a method like gradient descent is used
$\rightarrow$ Wouldn’t the *condition number* of the Hessian (e.g. via the ratio of max/min Eigenvalues) also play a big role? I’d like to hear the authors’ thoughts on this, and to what extent this is already factored into their approach, and whether they could provide additional insights.

- I’m wondering whether the authors could provide some further insights to what extent their theoretical insights based on the simplified 2-modal distribution (GMM) would generalise (or could be generalised) to more complex real-world distributions; I can see some additional information provided in appendices B.2 – B.4, but it seems to mostly discuss what aspects other works have or haven’t covered yet;  I understand that simplified models as well as assumptions are required for theoretical analyses, but it might be interesting for the wider audience to have some insights/comments on how the authors see the remaining gap between their findings and current models used in practice

**Minor / Recommendations:**
- The authors often use the expression ‘large optimization steps’ (e.g. abstract l. 029/030); To me, this seems more related to the optimizer needing to ‘move far’ to get to the solution, which I think would be better expressed by “many steps” (if the authors want to use ‘steps’), given that most gradient-decent-like optimisers operate with a fixed step size (albeit rescaled internally);
- Recommendation: The authors repeatedly state that “the model can not learn the *ground-truth* parameters” (e.g. l.024, l.084/085, etc.). While I understand that for small toy’ish theoretical setups, a ground-truth set of parameters exists (e.g. via close-form solution or other assumptions), this formulation can be quite confusing for the wider ML audience, given that for the actual ‘practical’ deep-learning-based setting that the authors address here, there usually is no ‘one ground-truth’ solution for the parameters.
$\rightarrow$ I’d hence recommend the authors to adapt the wording around these statements, e.g. to ‘optimal set of parameters’ / 'near-optimal set of parameters', or sth along these lines.

---

> ### Author Response · Authors · 2025-12-03
>
> **Weakness 1 && Q1: Influence of different sampling process**
>
> Thanks for the constructive discussion on the stochastic and deterministic sampling process. Assuming facing the same bad pretrained models (the same estimation and optimization error, named $\epsilon*{\text{score}}$), the performance is determined by the inference method (reverse SDE or reverse PFODE). As shown in the analysis of sampling (inference complexity) [1] [2], the stochasticity of reverse SDE plays an important role in alleviating error accumulation and only introduces an additional $\text{poly}(\epsilon*{\text{score}})$ in the sampling process. On the contrary, in the pure PFODE process (without stochastic corrector), as shown in [2], there will be an exponential term $\exp(\epsilon_{\text{score}})$ in the sampling analysis. Hence, facing the same bad pretrained score, the PFODE suffers from a worse sample quality compared to reverse SDE (our real-world experiments also support this discussion).
>
>
> **Weakness 2 && Q2：Insights**
>
> We thank the reviewer for connecting our work to the broader few-shot learning literature. We agree that the empirical result "bad models adapt worse" is intuitive. However, we respectfully argue that our contribution lies in **formalizing this intuition specifically for Diffusion Models**, where the optimization mechanism differs fundamentally from standard FSL.
>
> While the *phenomenon* is known, the *mathematical cause* in the context of **Score Matching** has been unexplored.Unlike standard FSL (typically Cross-Entropy or Metric learning), diffusion models optimize a score matching objective.And we provide the first theoretical proof of diffusion models.
>
> In addition, our main goal is to deepen our understanding of the phenomenon that few-shot diffusion models are optimized quickly.We completely agree that convexity alone guarantees only eventual convergence, and the **condition number** (ratio of max/min eigenvalues) is the decisive factor for convergence *speed* ("few steps").
>
> **Our theoretical analysis explicitly factors this in.** The convergence guarantee in **Theorem 6.4**  is derived directly from the condition number $\kappa$:
> $$\kappa = \frac{\alpha+\gamma+\zeta}{\alpha-\gamma} \approx \frac{\lambda_{max}}{\lambda_{min}}$$
> where the numerator corresponds to the Lipschitz constant $L_m$ (an upper bound of $\lambda_{max}$) and the denominator corresponds to the strong convexity parameter $\mu$ (a lower bound of $\lambda_{min}$).
>
>
>
> **Weakness 3 && Q3: General K-mode latent**
>
> We thank the reviewer for the valuable comments, and based on the reviewer's insightful feedback, we have successfully extended our proofs to the **general K-mode setting**.We provide a detailed K-mode analysis in the global response now.
>
> **Minor Weakness/ Recommendations**
>
> We thank the reviewer for the valuable comments, we will modidy them in the next version.
>
> [1] Chen, Sitan, et al. "The probability flow ode is provably fast." *Advances in Neural Information Processing Systems* 36 (2023): 68552-68575.
>
> [2]Yang, Ruofeng, et al. "Leveraging drift to improve sample complexity of variance exploding diffusion models." *Advances in Neural Information Processing Systems* 38 (2024): 107662-107702.

---

### Official Review · Reviewer_sceD · 2025-11-02

**Soundness:** 3
**Presentation:** 3
**Contribution:** 3
**Rating:** 6
**Confidence:** 4

**Summary:**

This paper investigates the optimization process of few-shot diffusion models. It addresses a critical gap in existing literature by focusing on why few-shot models require only a small number of optimization steps to achieve high performance, rather than merely explaining the sufficiency of limited data. This paper combines empirical experiments with theoretical analysis to demonstrate that the quality of pre-trained models is pivotal: great pre-trained models warm up the few-shot phase, leading to a strongly convex landscape and fast convergence, while bad pre-trained models (overfitting or underfitting) cause failures such as memory phenomena or loss gaps. The paper provides the first convergence guarantee for few-shot diffusion models under simplified assumptions.

**Strengths:**

1. The paper proves that great pre-trained models induce strong convexity in the few-shot objective function, leading to linear convergence rates for gradient descent (Theorem 6.4). This is the first convergence analysis for few-shot diffusion models, offering a solid foundation for future work.
2. This paper provides a new theoretical lens for understanding few-shot diffusion models, moving beyond data efficiency to computational efficiency.
3. This paper conducts experiments using standard datasets and models to validate theoretical claims.

**Weaknesses:**

1. The theoretical analysis is built on strong simplifications (e.g., linear subspaces, 2-mode GMMs), and the extension to K-mode GMMs in Appendix B remains intuitive rather than rigorous. Nonetheless, the empirical support and conceptual value of this discussion are sufficient to prevent this limitation from affecting my overall score.
2. The "bad" pre-trained models in the paper are created through extreme and artificial operations, such as overfitting Stable Diffusion 3 (SD3) Medium on only 5 dog images for 1,000 steps. This approach does not reflect real-world failure modes, where bad models typically arise from poor data quality, suboptimal architecture choices, or inadequate training strategies, rather than targeted overfitting.
3. The paper should establish evaluation metrics that are independent of fine-tuning outcomes (e.g., representation smoothness or distribution distance), and through large-scale experiments, quantify the correlation between pre-trained model metrics (such as FID) and few-shot performance, thereby providing actionable predictive guidelines.

**Questions:**

This paper presents a pioneering theoretical contribution by systematically analyzing the optimization process of few-shot diffusion models, demonstrating for the first time that high-quality pre-trained models induce a strongly convex landscape—enabling fast convergence with rigorous guarantees. Although limitations like the absence of quantifiable pre-training evaluation criteria and simplified theoretical assumptions slightly narrow its immediate applicability, these do not undermine the core innovation, as the work successfully bridges theory and practice, offering valuable insights for rapid customization in generative AI and setting a foundation for future research.

---

> ### Author Response · Authors · 2025-12-03
>
> **Weakness 1: Strong Simplifications**
>
> We thank the reviewers for their valuable comments, and based on the reviewer's insightful feedback, we have successfully extended our proofs to the **general K-mode setting**.We provide a detailed K-mode analysis in the global response now.
>
> **Weakness 2: How to get "Bad" pre-trained models**
>
> We thank the reviewers for their valuable comments. We agree that in practical applications, "bad" models are usually due to poor data quality or insufficient training, rather than intentional overfitting.
>
> However, we want to clarify that the "dog image overfitting" experiment was intended as an **extreme** simulation to visualize the theoretical concept of the "memory phenomenon" discussed in Section 4. We experimentally validated our theoretical findings: biased initialization may create a strong local minimum that is difficult to escape during fine-tuning with few samples, leading to failure in learning new concepts (e.g., cats). This can serve as an explanation for failed fine-tuning cases in reality.
>
> **Weakness 3: Metrics**
>
> For quantitative metrics, we consider a diffusion model to generate high-quality samples if the FID is less than 5 (where a large-scale open source diffusion model satisfies this requirement). However, for the generalization property, it is more difficult to measure this property for diffusion models. As shown in [1], a training interval exists between the memorization and generalization regimes, and we can employ the early stopping technique to avoid the memorization regime. An actionable strategy is to evaluate FID and generalization score [2]  during the training phase (e.g., every 100 epochs), and perform early stopping when the Generating score increases and the FID remains low.
>
> [1] Why Diffusion Models Don’t Memorize: The Role of Implicit Dynamical Regularization in Training. Announcing the NeurIPS 2025 Best Paper
>
> [2] Li, Xiang, Yixiang Dai, and Qing Qu. "Understanding generalizability of diffusion models requires rethinking the hidden gaussian structure." Advances in neural information processing systems 38 (2023): 57499-57538.

---

### Author Response · Authors · 2025-12-03
**K-mode GMM Derivation**

## 1. Score Function Definition

For a K-mode GMM, the score function $s_\theta(x)$ is given by:

$$
s_\theta(x) = {V \left( \sum_{i=1}^K \omega_{i,t} \hat{\mu}_{i,t} \right)} +{\left(\frac{1}{\sigma_t^2} - 1\right) V V^\top x}- \frac{1}{\sigma_t^2} x
$$

We still analyze **Squared Term** and **Cross Term** separately, and provide explicit expressions of Convexity bounds.

Let $z = V^\top x$ ,  $\bar{\mu} = \sum_{i=1}^K \omega_{i,t} \hat{\mu}_{i,t}$,

$\omega_{i,t} = \frac{e_{i,t}}{\sum_{k=1}^K e_{k,t}}$
and   $e_{i,t} = \exp\left(-\frac{1}{2}(z - \hat{\mu}_{i,t})^2\right)$.

## 2. Calculating the Squared Term


The full Jacobian matrix $M$ is:


$$
M = \\underbrace{\left[ \bar{\mu} + \left(\frac{1}{\sigma_t^2} - 1\right) (V^\top x) \right]}\_{\\alpha} I_D + \\underbrace{\left[ \sigma_{var}^2 + \left(\frac{1}{\sigma_t^2} - 1\right) \right]}\_{\\beta} V x^\top
$$


The Squared Term of the Hessian is approximated by $N = M M^\top$.
The matrix $M$ is in the exact form required by **Lemma F.1**:
$$
M = \alpha I + \beta xy^\top \quad (\text{Here } y = V)
$$

Using the closed-form solution for eigenvalues from Lemma F.1, the minimum eigenvalue is:

$$
\lambda_{\min}(N) = \min \left( \alpha^2, \quad \alpha^2 + \alpha \beta (V^\top x) + \frac{\beta^2 \|V\|^2 \|x\|^2}{2} - \frac{|\beta|}{2} \sqrt{\mathcal{D}} \right)
$$

Where discriminant $\mathcal{D}$:
$$
\mathcal{D} = \|V\|^2 \|x\|^2 \left( 4\alpha^2 + 4\alpha \beta (V^\top x) + \beta^2 \|V\|^2 \|x\|^2 \right)
$$



## 3. Cross Term

We analyze the cross term. Let $y(x) = s_\theta(x) - s_{target}(x)$ to quantify the impact of the estimation error $\Delta = V - A_{target}$ on the optimization landscape. We then derive a strict upper bound for the contribution to the critical matrix $M(x)$.

The derivation proceeds in three steps:**Bounding $y(x)$**,**Bounding the Matrix $M(x)$** and **Combining the Bounds**.


The final bound for the norm of $M(x)$, controlled by the initial parameter error $\|\Delta\|_2$:

$$
\|M(x)\|_2 \le \mathcal{P}(\|x\|_2) \cdot \|\Delta\|_2
$$

The controlling coefficient $\mathcal{P}(\|x\|_2)$ is a **3rd-order polynomial** in the state norm $\|x\|_2$. This implies that as the magnitude of the state $x$ (i.e., $\|x\|_2$) grows, the impact of the parameter error $\Delta$ on $M(x)$ grows cubically.


## 4. Final Expectation and Convexity Conclusion

To evaluate the Hessian over the entire dataset, we take the expectation with respect to the data distribution $p_{data}$ (K-mode GMM).

### 4.1 The Final Constant $C$
We define the **Expected Cross-Term Lipschitz Constant** $C_{final}$:

$$
C_{final} \triangleq \mathbb{E}\_{x \sim p_{data}} [\mathcal{P}(\|x\|_2)]
$$

Since the data follows a Gaussian Mixture Model, all moments $\mathbb{E}[\|x\|^n]$ (for $n=1,2,3$) are finite. Therefore, **$C_{final}$ is a finite positive constant.**

### 4.2 The Strong Convexity Condition
The minimum eigenvalue of the full Hessian is bounded by:

$$\lambda_{\min}(H) \ge 2 \lambda_{\min}(N) - 2 \\| \mathbb{E}[M] \\|_2$$

$$\lambda_{\min}(H) \ge 2 \alpha - 2 C_{final} \|\Delta\|_{init}$$

**Conclusion:**
Provided that the pre-training is "Great" (i.e., the initialization error $\|\Delta\|_{init}$ satisfies the condition below), the Hessian remains positive definite:

$$\|\Delta\|_{init} < \frac{\alpha}{C_{final}} \implies \lambda_{\min}(H) > 0$$

This proves that under K-mode GMM assumptions, a good initialization guarantees a **Strongly Convex** optimization landscape, ensuring fast convergence.

---

### Meta-Review · Area_Chair_t96R · 2026-01-05

**Summary:**

Reviewers raise serious concerns about the realism, necessity, and significance of the theoretical analysis. The guarantees rely on highly simplified and contrived assumptions (e.g., linear subspaces, low-dimensional Gaussian mixtures, and specific shallow score networks) that are far removed from real-world diffusion models and image data, casting doubt on their practical relevance. Moreover, the reviewers question whether the score-matching–based analysis provides fundamentally new insights beyond existing transfer learning theory, given the equivalence between diffusion training objectives and weighted MSE regression. Overall, the paper lacks convincing justification that its theoretical results meaningfully explain or generalize to large-scale diffusion models in practice, and the limited, artificial empirical validation does not sufficiently bridge this gap. Therefore, I recommend rejection.

**Reviewer Concerns:**

Reviewer sceD and Reviewer keJV 's concerns are addressed, Reviewer PsXu and Reviewer 458S's concerns are still outstanding.

**Reviewer Scores:**

I think Reviewer PsXu and Reviewer 458S could potentially raise their scores by 0–2 points.

---

### Decision · Program_Chairs · 2026-01-26

Reject